# Variational Inference for Interacting Particle Systems with Discrete Latent States

**Giosuè Migliorini**
Department of Statistics
University of California, Irvine

**Padhraic Smyth**
Department of Statistics and Computer Science
University of California, Irvine

## Abstract

We present a novel Bayesian learning framework for interacting particle systems with discrete latent states, addressing the challenge of inferring dynamics from partial, noisy observations. Our approach learns a variational posterior path measure by parameterizing the generator of the underlying continuous-time Markov chain. We formulate the problem as a multi-marginal Schrödinger bridge with aligned samples, employing a two-stage learning procedure. Our method incorporates an emission distribution for decoding latent states and uses a scalable variational approximation.

## 1 Introduction

Many real-world phenomena, from epidemics to wildfires, can be modeled as systems of interacting components evolving in continuous time, where the underlying dynamics are governed by discrete latent states. This paradigm extends the concept of hidden Markov models [Baum and Petrie, 1966, Kouemou and Dymarski, 2011] to spatially structured, continuous-time processes. Interacting particle systems (IPSs) [Liggett, 1985, Lanchier, 2024] offer a powerful mathematical framework for describing local propagation dynamics. However, inferring the rules governing these systems from partial, noisy observations remains a significant challenge. We propose a novel Bayesian approach that addresses this challenge by learning a variational posterior path measure on the space of IPS trajectories. Our approach parameterizes the rate matrix of the continuous-time Markov chain (CTMC) of the latent IPS using neural networks and incorporates an emission model that can decode internal discrete states to continuous data and noisy observations. Key contributions of our approach include:

- Framing the problem as a multi-marginal discrete Schrödinger bridge, solved by a two-stage procedure: learning an endpoint-conditioned process for trajectory reconstruction, followed by distillation to an unconditional process for prediction.

- A scalable variational approximation using site-wise factorization of time-marginals and assuming independent particle evolution in infinitesimal time intervals conditionally on the present global configuration, enabling efficient learning for high-dimensional spatio-temporal processes.

- Flexibility in incorporating domain knowledge through informative priors on rate matrix entries and neural architectures with desirable inductive biases.

We demonstrate preliminary results of our approach on two simulated datasets for the following tasks: reconstructing the trajectory of an epidemic on a network and predicting wildfire spread on a lattice. For a description of the notation, see Appendix A. An overview of the relevant literature is presented in Appendix B, while proofs and other derivations are provided in Appendix C.

Workshop on Bayesian Decision-making and Uncertainty, 38th Conference on Neural Information Processing Systems (NeurIPS 2024).

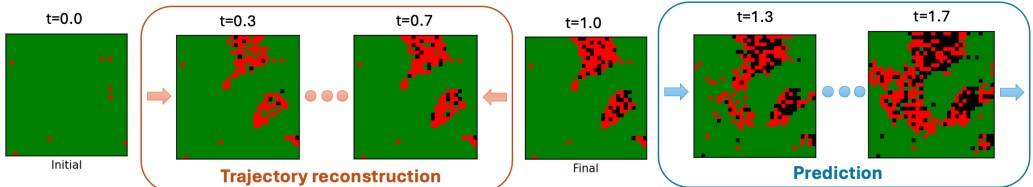

Figure 1: An illustration of our methodology on a simulated noiseless dataset of wildfire propagation. The first model approximates a Markov bridge interpolating between the observations, enabling to reconstruct the unobserved trajectory. The second model, approximating the unconditional process, can predict beyond the last observation. Results shown for a held-out example.

## 2 Background

**Interacting particle systems**   Consider a graph $\mathcal{G} = (V, E)$, and denote $i \sim j$ if there is an edge between the vertices $i, j$, i.e., $\{i, j\} \in E$. Following Liggett [1985], we refer to vertices $i \in V$ as sites. For a countable local state space $S$, consider the configuration space $\mathcal{Z} := \{z \mid z : V \to S\}$. For our analysis, we assume both $V$ and $S$ to be finite. An IPS adds a continuous-time dimension to this setting. Specifically, we obtain a CTMC $z(t)$ on $\mathcal{Z}$ restricted to a time interval $[0, T]$, whose path space we denote $\Omega_{[0,T]}$. We define $z^i(t) \in S$ as the state of site $i$ at time $t$. We consider a scenario where the dynamics of each site are described by local transition rates that depend on the graph's connectivity [Lanchier, 2017], corresponding to

$$\lambda_t^{s \to \tilde{s}}(i, z(t)) := \lim_{\Delta t \to 0} \frac{1}{\Delta t} \mathbb{P}\left(z^i(t + \Delta t) = \tilde{s} \mid z^i(t) = s, z^j(t) : i \sim j\right),$$

for $s$ to $s' \neq s$ at site $i$ and time $t \in [0, T]$, and set $\lambda_t^{s \to s}(i, z(t)) := -\sum_{s' \neq s} \lambda_t^{s \to s'}(i, z(t))$. The local transition rates for each site can be compactly represented as matrices $\boldsymbol{\lambda}_t(i, z(t))$.

**Definition 1** (Local generator). *A mapping $\Lambda_t : \mathcal{Z} \times [0, T] \to \mathbb{R}^{|V| \times |S| \times |S|}$ assigning to each configuration $z(t)$ a three-dimensional array containing the local transition rate matrices $\boldsymbol{\lambda}_t(i, z(t))$ for all sites $i \in V$.*

One can characterize the CTMC on the space of configurations by making the additional assumption that updates at each site happen independently from one another. Then, for an arbitrarily small $\Delta t$ and $\tilde{z} \in \mathcal{Z}$,

$$p_{t+\Delta t|t}(\tilde{z}|z) \approx \delta_{z,\tilde{z}} + \Delta t \sum_{i \in V} \lambda_t^{z^i \to \tilde{z}^i}(i, z(t)) \prod_{j \neq i} \delta_{z^j, \tilde{z}^j} + o(\Delta t). \tag{1}$$

For brevity, we denote these transition rates as $\Lambda_t(\tilde{z} \mid z) := \sum_{i \in V} \lambda_t^{z^i \to \tilde{z}^i}(i, z(t)) \prod_{j \neq i} \delta_{z^j, \tilde{z}^j}$. A detailed derivation can be found in Appendix C.3. We refer to endpoint-conditioned processes as Markov bridges, and we provide a quick overview in Appendix C.1 for noisy data.

## 3 Variational Discrete Interacting Particle Systems

We consider a dataset of sequences of observations in a space $\mathcal{X}$ and observation times $\{x_{1:K_j}^{(j)}, t_{1:K_j}^{(j)}\}_{j=1:N}$. We assume these are noisy observations of a latent IPS $(z^{(j)}(t))_{t \in [t_1^{(j)}, t_{K_j}^{(j)}]} \in \Omega_{[t_1^{(j)}, t_K^{(j)}]}$. Pairwise conditional independence is assumed for any couple of observations in a sequence, i.e. $x_k^{(j)} \perp\!\!\!\perp x_{\tilde{k}}^{(j)} \mid z^{(j)}(t)$ for $t \in [t_k^{(j)}, t_{\tilde{k}}^{(j)}]$ and $t_k^{(j)} < t_{\tilde{k}}^{(j)}$. The discrete set of measurement times $t_1^{(j)} < \cdots < t_{K_j}^{(j)}$ is allowed to be arbitrarily defined for each sequence, e.g., at random or regularly spaced. For ease of illustration, we present our results for a fixed set of observation times $t_1, \ldots, t_K$, but the extension to irregularly sampled time series is straightforward and presented in Appendix C.2. We assume that the graph determining the particles' dependence structure is fixed for each realization and directly deducible from the observed sequences.

Consider an emission distribution $p_t(x \mid z) \in \mathcal{P}(\mathcal{X})$ and a prior path measure $P \in \mathcal{P}(\Omega_{[t_1, t_K]})$ for the latent IPS. This can be specified directly on the entries of a prior local generator, encoding possible constraints in the latent dynamics, and by an initial prior distribution. Let $\mathbb{P} \in \mathcal{P}(\mathcal{X}^K \times \Omega_{[t_1, t_K]})$

denote the reference measure constructed by gluing the prior and emission probabilities at each observed timestep, i.e. $\mathrm{P}(d\boldsymbol{x}_{1:K}, (d\boldsymbol{z}(t))_{t\in[t_1,t_K]}) = \prod_{k=1}^{K} p_{t_k}(d\boldsymbol{x}_k \mid \boldsymbol{z}(t_k))P((d\boldsymbol{z}(t))_{t\in[t_1,t_K]})$. The marginal distribution of the data at an observation time $t_k$ is denoted as $\pi_k \in \mathcal{P}(\mathcal{X})$, for $k = 1, \ldots, K$. For a given sequence of distributions $\{\pi_k\}_{k=1:K}$, we can express a multi-marginal discrete Schrödinger bridge problem with noisy observations as

$$Q^* := \underset{Q \in \mathcal{P}(\mathcal{X}^K \times \Omega_{[t_1,t_K]})}{\arg\min} \{D_{\mathrm{KL}}(Q \,\|\, \mathrm{P}) \mid q_{t_k} = \pi_k, \, k = 1, \ldots, K\}, \tag{2}$$

where $q_{t_k} \in \mathcal{P}(\mathcal{X})$ correspond to marginalizations of Q at each observed timepoint in the space of observations $\mathcal{X}$.

Our goal is twofold:

- **Trajectory reconstruction**, by learning the conditional local generator $\boldsymbol{\Lambda}_t(\cdot \mid \boldsymbol{x}_{1:K})$ of the Markov bridge $Q^\star_{\cdot|\boldsymbol{x}_{1:K}} \in \mathcal{P}(\Omega_{[t_1,t_K]})$;
- **Prediction**, by learning the local generator $\boldsymbol{\Lambda}_t$ of the Markov process $Q^\star \in \mathcal{P}(\Omega_{[t_1,t_K]})$, enabling extrapolation beyond an observed time window or with no past observations at all for a given graph.

We show that the second goal can be achieved by distilling knowledge from a model trained for the first goal into a model that does not glance at future observations.

### 3.1 Trajectory reconstruction

Let $\pi_{1:K}$ denote the coupling solving the static version of (2), that is

$$\pi_{1:K} = \underset{q_{1:K} \in \mathcal{P}(\mathcal{X}^K)}{\arg\min} \{D_{\mathrm{KL}}(q_{t_{1:K}} \,\|\, p_{t_{1:K}}) \mid q_k = \pi_k, \, k = 1, \ldots, K\}, \tag{3}$$

where $p_{1:K} \in \mathcal{P}(\mathcal{X}^K)$ is the marginal of the observed trajectories obtained from the reference measure P. Similarly to the setting considered in Somnath et al. [2023], we assume that our dataset is comprised of trajectories of *aligned* samples, in the sense that each observed trajectory $\boldsymbol{x}_{1:K}$ is sampled from the coupling $\pi_{1:K}$. By the additive property of the Kullback-Leibler divergence [Léonard, 2013], the dynamic problem in equation 2 can be rewritten as

$$\underset{Q \in \mathcal{P}(\Omega_{[t_1,t_K]})}{\arg\min} \mathbb{E}_{\pi_{1:K}} \left[ D_{\mathrm{KL}}(Q_{\cdot|\boldsymbol{x}_{1:K}} \,\|\, P_{\cdot|\boldsymbol{x}_{1:K}}) \right]. \tag{4}$$

As samples from $\pi_{1:K}$ are available, we can treat this stage as a *smoothing* problem, and perform approximate posterior inference.

#### 3.1.1 Noiseless data

In the special case where observations are noiseless snapshots of the IPS, i.e. $\boldsymbol{x}_k = \boldsymbol{z}(t_k)$, the latent variables in the model correspond to the unobserved portions of the stochastic process of the form $(\boldsymbol{z}(t))_{t\in(t_k,t_{k+1})}$. The emission distribution corresponds to the transition probability $p_{t_k}(\boldsymbol{x} \mid \boldsymbol{z}) = \lim_{t \to t_{k+1}^-} \mathbb{P}(\boldsymbol{z}(t_k) = \boldsymbol{x} \mid \boldsymbol{z}(t) = \boldsymbol{z})$, obtained from the prior rates using equation 1. We learn a variational posterior $Q^\theta \in \mathcal{P}(\Omega_{[t_1,t_K]})$ through amortization [Amos et al., 2023], by parameterizing the local generator of the approximate Markov bridge with a neural model $\Lambda^\theta$, having parameters $\theta \in \Theta$.

**Proposition 2.** *Let (3) admit a solution $\pi_{1:K}$. Moreover, let $\boldsymbol{x}_{1:K}$ be noiseless observations of $(\boldsymbol{z}(t)) \in \Omega_{[0,T]}$, and let $P \in \mathcal{P}(\Omega_{[0,T]})$. Then, the amortized version of the problem in equation 2 reduces to*

$$\underset{\theta \in \Theta}{\arg\min} \sum_{k=1}^{K-1} \mathbb{E}_{\pi_{k,k+1}} \left[ D_{\mathrm{KL}}(Q^\theta_{\cdot|\boldsymbol{x}_k,\boldsymbol{x}_{k+1}} \,\|\, P) - \mathbb{E}_{Q^\theta_{\cdot|\boldsymbol{x}_k,\boldsymbol{x}_{k+1}}}[\log p_{t_{k+1}}(\boldsymbol{x}_{k+1} \mid \boldsymbol{z}(t_{k+1}^-))] \right], \tag{5}$$

*where $\pi_{k,k+1} \in \mathcal{P}(\mathcal{X}^2)$ is obtained by marginalizing $\pi_{1:K}$, and $\boldsymbol{z}(t_{k+1}^-) = \lim_{t \to t_{k+1}^-} \boldsymbol{z}(t)$.*

Notice that this parameterization is highly scalable as it allows mini-batching across segments of time. The KL divergence of two CTMCs can be estimated using Monte Carlo integration, using the analytic form derived in Opper and Sanguinetti [2007], see Appendix C.4 for a derivation.

### 3.1.2 Noisy data

In order to learn a conditional model with noisy data, we propose to parameterize our variational posterior in an autoregressive fashion, extending the method proposed in Seifner and Sánchez [2023]. The authors propose to compute a single hidden representation of the entire sequence via an ODE-RNN model [Rubanova et al., 2019], and then condition the inference model at every time step using that variable. We extend their approach by letting the conditioning variable change through time, only capturing dependence on future observations. Note that the option to drop conditioning on past observations follows naturally from the conditional independence assumption. We do not need to train multiple models to accomplish this, as it is enough to checkpoint the ODE-RNN model at the observation times. We can express the variational posterior as

$$q_{t_1}^{\theta}(d\boldsymbol{z}(t_1) \mid h_{t_1}(\boldsymbol{x}_{1:K})) \prod_{k=1}^{K-1} dQ^{\theta}((d\boldsymbol{z}(t))_{t\in(t_k, t_{k+1}]} \mid \boldsymbol{z}(t_k), h_{t_k}^{\theta}(\boldsymbol{x}_{k+1:K})), \qquad (6)$$

where $q_{t_1}^{\theta}$ is a Categorical distribution parameterized by an encoder. The model can be learned by minimizing the negative evidence lower bound

$$\mathcal{L}^{\mathrm{AR}}(\theta) := \mathbb{E}_{\pi_{1:K}}\left[D_{\mathrm{KL}}(Q_{\cdot|\boldsymbol{x}_{1:K}}^{\theta} \,\|\, P) - \mathbb{E}_{Q_{\cdot|\boldsymbol{x}_{1:K}}^{\theta}}\left[\sum_{k=1}^{K}\log p_{t_k}(\boldsymbol{x}_k \mid \boldsymbol{z}(t_k))\right]\right]. \qquad (7)$$

### 3.1.3 Simulation

While at sampling time any exact stochastic simulation algorithm (e.g. Gillespie 2001) can be employed, at training time we are limited to differentiable approximations. We propose two options, trading off assumptions on the variational family for scalability.

**Forward simulation** This approach involves fixing a time-discretization grid $t_k < t_k + \Delta t < \cdots < t_{k+1} - \Delta t < t_{k+1}$ and sampling iteratively from a Gumbell-softmax approximation [Jang et al., 2017] to equation 1, updating the latent state $\boldsymbol{z}(t + \Delta t) = \boldsymbol{z}(t) + N_t^{\theta}(\Delta t, \boldsymbol{z}(t))$, where $N_t^{\theta}$ is the jump process describing the latent CTMC. While this method is exact in the limit $\Delta t \to 0$ and requires no additional restrictions to the variational family, its cost scales linearly with respect to the number of jumps [Jia and Benson, 2019]. However, we are not required to compute inflow rates (of the form $\lambda^{s \to z^i}$), but only outflow rates (like $\lambda^{z^i \to s}$), making the output of our local rates model scale linearly with respect to $|S|$.

**Neural master equation** Techniques from the literature on neural ODEs [Chen et al., 2021] can be applied if we consider a factorized posterior $q_t(\boldsymbol{z} \mid \boldsymbol{x}_{1:K}) = \prod_{i\in V} q_t^i(z^i \mid \boldsymbol{x}_{1:K})$. Note that spatial dependence is still propagated through time, as the local rates model depends on the global configuration (or a neighborhood restriction). For notational simplicity we omit conditioning on $\boldsymbol{x}_{1:K}$, but note that this applies to conditional and unconditional settings alike. We can then simulate from the system of marginal master equations given initial conditions $q_1^i(z_1^i)$, $i \in V$, as

$$\partial_t q_t^i(z^i(t)) = \sum_{s\neq z^i(t)}\left(\mathbb{E}_{q_t^{-i}}\left[\lambda_t^{s\to z^i(t)}(i, \boldsymbol{z}(t))\right]q_t^i(s) - \mathbb{E}_{q_t^{-i}}\left[\lambda_t^{z^i(t)\to s}(i, \boldsymbol{z}(t))\right]q_t^i(z^i(t))\right), \ i \in V.$$
$$(8)$$

This variational approximation was introduced for continuous-time Bayesian networks in Linzner and Koeppl [2018] under the name of *star*-approximation. This is to be distinguished from the *mean-field* approach, where the approximation entails either a fixed rate for each site or compartmental models directly describing the mean-field behaviour of the system [Seifner and Sánchez, 2023, Opper and Sanguinetti, 2007, Cohn et al., 2010]. As the solution to equation 8 is a continuous function, one can use the memory-efficient adjoint method [Chen et al., 2021, Seifner and Sánchez, 2023] at training time, making this approach extremely scalable.

## 3.2 Prediction

The trajectory reconstruction model learned in Section 3.1 approximates the Schrödinger bridge $Q^{\star}$ through an endpoint-conditioned scheme for the latent trajectories, leveraging the factorization $Q_{\cdot|\boldsymbol{x}_{1:K}}^{\theta}((d\boldsymbol{z}(t))_{t\in[t_1, t_K]})\pi_{1:K}(d\boldsymbol{x}_{1:K})$. However, for many applications, we require the ability to

generate predictions beyond observed time intervals. Given an initial observation $\boldsymbol{x}_1$ at time $t_1$, we aim to predict observations at arbitrary times $\tilde{t} \in (t_1, t_K]$. This prediction task leverages an alternative factorization of $Q^\star$:

$$q_{\tilde{t}}^\star(d\boldsymbol{x}_{\tilde{t}} \mid \boldsymbol{z}(\tilde{t}))Q_{\cdot|\boldsymbol{x}_1}^\star((dz(t))_{t \in [t_1, t_K]})\pi_1(d\boldsymbol{x}_1).$$

While it can be shown that $q_{\tilde{t}}^\star(\boldsymbol{x}_{\tilde{t}} \mid \boldsymbol{z}(\tilde{t})) = p_{\tilde{t}}(\boldsymbol{x}_{\tilde{t}} \mid \boldsymbol{z}(\tilde{t}))$ using the additive property of the KL, the models developed thus far are constrained by their dependence on endpoint conditions. To overcome this limitation, we propose learning an unconditional amortized posterior $Q^\phi$ by minimizing the KL divergence

$$\mathcal{L}_{\mathrm{KL}}(\phi) \coloneqq D_{\mathrm{KL}}(Q^\star \,\|\, Q^\phi) \propto \mathbb{E}_{\pi_{1:K}}\left[D_{\mathrm{KL}}(Q_{\cdot|\boldsymbol{x}_{1:K}}^\star, \|, Q^\phi)\right]. \tag{9}$$

A direct computation of this loss is intractable due to the unavailability of $Q^\star$ and $Q_{\cdot|\boldsymbol{x}_{1:K}}^\star$, hence we employ the surrogate loss function

$$\hat{\mathcal{L}}_{\mathrm{KL}}^\theta(\phi) \coloneqq \mathbb{E}_{\pi_{1:K}}\left[D_{\mathrm{KL}}(Q_{\cdot|\boldsymbol{x}_{1:K}}^\theta, \|, Q^\phi)\right]. \tag{10}$$

The absolute difference between these quantities can be upper bounded in terms of the total variation distance between the solution to equation 2 and our conditional approximation. We provide a detailed analysis of the bound in Appendix C.6.

## 4  Experiments

We demonstrate our methodology on two simulated scenarios: epidemic trajectory inference on networks and wildfire spread prediction on lattices. We parameterize the neural models for the local generators with a novel architecture, detailed in Appendix D. Results and details of the simulations are reported in Appendix E.

## 5  Conclusion

We introduce a variational inference method to fit partially observed trajectories whose dynamics can be modeled by a continuous-time latent process, parameterized as an interacting particle system. Our solution is an approximation to a multi-marginal Schrödinger bridge, that we obtain by first fitting an endpoint-conditioned model and then distilling it into an unconditional one. This methodology enables both trajectory reconstruction and prediction of future states. In future work we aim at testing our models on real data, comparing with state-of-the-art methods.

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

# A  Notation

Let $\Omega_{[0,T]}$ be the space of $\mathcal{Z}$-valued cadlag functions over a time interval $[0,T]$, and denote by $\mathcal{P}(\Omega_{[0,T]})$ the space of probability measure on the path space. We denote by $\Omega_{[t,t']}$ time restrictions of $\Omega_{[0,T]}$ to $[t,t']$, for $0 \le t < t' \le T$. We denote the cartesian product $\times_{k \in [K]} \mathcal{X}$ of observations at $K$ times as $\mathcal{X}^K$. Consider the Polish space $\mathcal{Q} := \mathcal{X}^K \times \Omega_{[0,T]}$ and probability measures $\mathrm{Q}, \mathrm{P} \in \mathcal{P}(\mathcal{Q})$. We introduce the following notation:

- The marginal probability measures over observations, given by the canonical projection $\phi : \mathcal{Q} \to \mathcal{X}^K$ and denoted as $q_{1:K} := \phi_\# \mathrm{Q}$, $p_{1:K} := \phi_\# \mathrm{P}$.

- The marginal path measures over latent trajectories, given by the canonical projection $\varphi : \mathcal{Q} \to \Omega_{[0,T]}$ and denoted as $Q := \varphi_\# \mathrm{Q}$, $P := \varphi_\# \mathrm{P}$.

- The conditional path measures over latent trajectories, given by measurable mappings $\boldsymbol{x}_{1:K} \in \mathcal{X}^K \mapsto Q_{\cdot | \boldsymbol{x}_{1:K}} \in \mathcal{P}(\Omega_{[0,T]})$ and $\boldsymbol{x}_{1:K} \in \mathcal{X}^K \mapsto P_{\cdot | \boldsymbol{x}_{1:K}} \in \mathcal{P}(\Omega_{[0,T]})$

For a path measure $Q \in \mathcal{P}(\Omega_{[0,T]})$, we assume that the time-marginal and transition probability measures are absolutely continuous w.r.t. the counting measure. Their Radon-Nikodym derivative can then be expressed by the probability mass function $q_t(\boldsymbol{z})$ and the transition probability $q_{t'|t}(\tilde{\boldsymbol{z}} \mid \boldsymbol{z})$ for timesteps $0 \le t < t' \le T$ and configurations $\boldsymbol{z}, \tilde{\boldsymbol{z}} \in \mathcal{Z}$.

# B  Related work

## B.1  Learning interacting particle systems

The dynamics of many physical systems can be described through the local interaction laws of their constituent components. This principle has inspired computational frameworks that directly parameterize these governing interactions, both deterministically and stochastically. A prime example is cellular automata [Wolfram, 1986, Grinstein et al., 1985]. Early developments focused on studying the emergence of global patterns from a fixed set of rules on the evolution of individual cells. The inverse problem —inferring such rules from observations— has been of historical interest in the machine learning community [Wulff and Hertz, 1992, Mordvintsev et al., 2020], with recent developments incorporating attention-based architectures, graph neural networks, and black-box variational inference [Tesfaldet et al., 2022, Kang et al., 2024, Grattarola et al., 2021, Palm et al., 2022]. Models that learn interaction rules find applications across many domains, including physical simulators, multi-agent dynamics, dynamic graphs, as well as deep generative modeling [Kalkhof et al., 2024].

Within this context, most existing methods have proposed iterative updating schemes by parameterizing transition rules in discrete time. Interacting particle systems (IPSs) offer an alternative mathematical formalism that extends cellular automata to continuous time. Interacting particle systems are structured CTMCs whose states evolve with dependence on neighbors within a topology, typically established by a graph. Lanchier [2017] provides a modern introduction to this field. Classical literature focused on systems with finite states and often countably many sites [Bramson and Griffeath, 1980, Liggett, 1985, Durrett, 2006], while more recent work has focused on systems with finitely many sites [Aldous, 2013]. These systems have found applications in multi-agent modeling [Comas et al., 2023] and have been extended to systems of stochastic differential equations (SDEs) in Euclidean space. This extension has seen increased attention recently [Lu et al., 2021, Yang et al., 2022, Feng et al., 2022, Liu et al., 2023, Lang et al., 2024, Kümmerle et al., 2024, Boffi and Vanden-Eijnden, 2024]. The learnability and identifiability of interaction rules in these systems have also been explored [Bongini et al., 2017, Li et al., 2021].

## B.2  Inference for CTMCs

Inference methods for Markov jump processes (MJPs) have been extensively studied. Maximum likelihood estimation for time-homogeneous MJPs is discussed in Jackson [2011], Bladt and Sørensen [2005], McGibbon and Pande [2015]. Expectation-maximization techniques for continuous-time hidden Markov models have been developed in Liu et al. [2015], and an overview of the topic can be found in Wang [2021]. Bayesian approaches include Markov chain Monte Carlo

methods [Boys et al., 2008, Hobolth and Stone, 2009, Rao and Teh, 2013] and variational methods. The latter include mean-field [Opper and Sanguinetti, 2007, Cohn et al., 2010], moment-based methods [Wildner and Koeppl, 2019], combinations with MCMC [Zhang et al., 2017], and extensions to hybrid processes [Köhs et al., 2021]. Novel methods include black-box variational inference with neural networks [Seifner and Sánchez, 2023], foundation models [Berghaus et al., 2024], and expectation propagation [Alt and Koeppl, 2023]. Another directly related line of research focuses on simulation methods for Markov bridges, notably Corstanje et al. [2023], Corstanje and van der Meulen [2023] and Golightly and Sherlock [2019]. While less directly related, it's worth noting recent work discrete flow matching and diffusion methods [Campbell et al., 2022, Igashov et al., 2023, Lou et al., 2023, Campbell et al., 2024]. Concurrently to our work, a similar formulation of discrete Schrödinger bridges as CTMCs for two endpoint marginals contraints has been proposed in the context of discrete generative modelling by Kim et al. [2024].

## B.3  Trajectory Inference

Trajectory inference is a crucial component of our work, with connections to several recent developments. The Schrödinger bridge (SB) problem with multi-marginal constraints has been explored by Chen et al. [2019], Lavenant et al. [2021]. Recent advances in SB methods with a source and a target are presented in Vargas et al. [2021] and De Bortoli et al. [2021], with extensions to the multi-marginal setting by Shen et al. [2024]. Our approach shares similarities with Somnath et al. [2023], Shi et al. [2024], and Peluchetti [2023] in that it relies on samples from couplings solving the static SB problem. However, our methodology differs in that we learn the Markovian bridge and recover the unconditional path measure by distillation, rather than relying on closed-form endpoint-conditioned diffusions. The concept of Markov bridge by interpolation with a fictitious dynamic, as proposed by Igashov et al. [2023], is related to stochastic interpolants [Albergo and Vanden-Eijnden, 2022, Tong et al., 2023, Lipman et al., 2022, Liu et al., 2022] for probabilistic forecasting [Chen et al., 2024]. Ad-hoc variants for dynamical systems have also been developed [Rühling Cachay et al., 2024]. Our methodology also shares connections with flow matching using Gaussian process and Kalman filter interpolants [Tamir et al., 2023], in the fact that we are interested in *model-based* interpolants in a Bayesian framework.

## C  Proofs

### C.1  Markov bridges

Consider a sequence of observations $\{\boldsymbol{x}_k\}_{k\in[K]} \in \mathcal{X}^K$ recorded at times $\{t_k\}_{k\in[K]} \in \mathbb{R}^K$, and assume conditional independence with respect to a Markov process $(\boldsymbol{z}(t))_{t\in[0,T]}$. For $t \in [t_0, t_{K-1}]$, let $\boldsymbol{x}_{>t} = \{\boldsymbol{x}_k \,|\, t_k > t,\, k = 1, \ldots, K\}$ and $\boldsymbol{x}_{\leq t} = \{\boldsymbol{x}_k \,|\, t_k \leq t,\, k = 1, \ldots, K\}$. The next observation after $t$ is at time $t' \coloneqq \min\{t_k : t_k > t,\, k = 1, \ldots, K\}$, and we assume $t + \Delta t < t'$ for $\Delta t \approx 0$, by right-continuity of the transition probabilities. We can then denote the conditional transition rates for $\tilde{\boldsymbol{z}} \neq \boldsymbol{z}$ as

$$
\begin{aligned}
\Lambda_t(\tilde{\boldsymbol{z}} \mid \boldsymbol{z},\, \boldsymbol{x}_{0:K}) &= \lim_{\Delta t \downarrow 0} (\Delta t)^{-1} \left[ \mathbb{P}(\boldsymbol{z}(t + \Delta t) = \tilde{\boldsymbol{z}} \mid \boldsymbol{z}(t) = \boldsymbol{z},\, \boldsymbol{x}_{0:K}) \right] \\
&= \lim_{\Delta t \downarrow 0} (\Delta t)^{-1} \left[ \frac{\mathbb{P}(\boldsymbol{z}(t + \Delta t) = \tilde{\boldsymbol{z}}, \boldsymbol{z}(t) = \boldsymbol{z},\, \boldsymbol{x}_{>t} \mid \boldsymbol{x}_{\leq t})}{\mathbb{P}(\boldsymbol{z}(t) = \boldsymbol{z},\, \boldsymbol{x}_{>t} \mid \boldsymbol{x}_{\leq t})} \right] \\
&= \lim_{\Delta t \downarrow 0} (\Delta t)^{-1} \left[ \frac{\mathbb{P}(\boldsymbol{x}_{>t+\Delta t} \mid \boldsymbol{z}(t + \Delta t) = \tilde{\boldsymbol{z}})\mathbb{P}(\boldsymbol{z}(t + \Delta t) = \tilde{\boldsymbol{z}} \mid \boldsymbol{z}(t) = \boldsymbol{z})}{\mathbb{P}(\boldsymbol{x}_{>t} \mid \boldsymbol{z}(t) = \boldsymbol{z})} \right] \\
&= \Lambda_t(\tilde{\boldsymbol{z}} \mid \boldsymbol{z}) \frac{\mathbb{P}(\boldsymbol{x}_{>t} \mid \boldsymbol{z}(t) = \tilde{\boldsymbol{z}})}{\mathbb{P}(\boldsymbol{x}_{>t} \mid \boldsymbol{z}(t) = \boldsymbol{z})},
\end{aligned}
$$

and similarly

$$
\Lambda_t(\boldsymbol{z} \mid \boldsymbol{z},\, \boldsymbol{x}_{0:K}) = -\sum_{\tilde{\boldsymbol{z}} \neq \boldsymbol{z}} \Lambda_t(\tilde{\boldsymbol{z}} \mid \boldsymbol{z}) \frac{\mathbb{P}(\boldsymbol{x}_{>t} \mid \boldsymbol{z}(t) = \tilde{\boldsymbol{z}})}{\mathbb{P}(\boldsymbol{x}_{>t} \mid \boldsymbol{z}(t) = \boldsymbol{z})}. \tag{11}
$$

We refer the reader to Fitzsimmons et al. [1992] for a detailed construction.

## C.2 Irregularly sampled time series

The problem in equation 2 considers a fixed number of time steps $K$ and a set of observation times $t_1, \ldots, t_K$. In this section, we provide an extension to irregular an arbitrary observation times.

We assume that the number of timesteps is i.i.d. for each trajectory, and drawn from $K \sim p(K)$. The same goes for observation times, that are in turn drawn from $t_{1:K} \mid K \sim p(t_{1:K} \mid K)$, and $\boldsymbol{x}_{1:K} | t_{1:K} \sim \pi_{1:K}$. We do not model these probabilities, and instead express equation 2 as a solution in expectation, i.e.

$$Q^*_{\cdot | t_{1:K}} \coloneqq \underset{Q \in \mathcal{P}(\mathcal{X}^K \times \Omega_{[t_1, t_K]})}{\arg\min} \{D_{\mathrm{KL}}(Q \,\|\, P) \mid q_{t_k} = \pi_k, \, k = 1, \ldots, K\}, \quad Q^* = \mathbb{E}_{K, t_{1:K}}\left[Q^*_{\cdot | t_{1:K}}\right]. \tag{12}$$

The amortized problem can be written as

$$\underset{\theta \in \Theta}{\arg\min} \, \mathbb{E}_{K, t_{1:K}}\{D_{\mathrm{KL}}(Q^\theta \,\|\, P) \mid q_{t_k} = \pi_k, \, k = 1, \ldots, K\}, \tag{13}$$

where $Q^\theta, P \in \mathcal{P}(\mathcal{X}^K \times \Omega_{[t_1, t_K]})$.

## C.3 From local interactions to a global dynamics

Consider a stochastic process $(\boldsymbol{z}(t)) \in \Omega_{[0,T]}$, whose dynamics at each site are driven by a system of CTMCs $(z^i(t))$, for $i \in V$. In this section we illustrate that, under an independence assumption of jumps in infinitesimal time intervals, a global description of the dynamics can be deduced. This corresponds to an CTMC on the global state space $\mathcal{Z} \coloneqq S^V$, hence a global master equation (ME) can be derived. This equivalence is well-known in the literature on continuous-time Bayesian networks [Nodelman et al., 2002, Linzner, 2021].

**Local dynamics.** Let $\tilde{\boldsymbol{z}}^{i,s} \in \mathcal{Z}$ be $\boldsymbol{z} \in \mathcal{Z}$ where we substitute site $i \in V$ to be $s \in S$, and denote

$$p_t^{i|-i}(s \mid \boldsymbol{z}) \coloneqq p_t^i\left(z^i(t) = s \mid \{\boldsymbol{z}_t(j) = \boldsymbol{z}(j), j \neq i\}\right),$$
$$p_{t+\Delta t|t}^i(s \mid \boldsymbol{z}) \coloneqq p_t^i\left(\boldsymbol{z}_{t+\Delta t}(i) = s \mid \{\boldsymbol{z}_t = \boldsymbol{z}\}\right),$$
$$p_t^{-i}(\boldsymbol{z}) \coloneqq \sum_{s \in S} p_t(\tilde{\boldsymbol{z}}^{i,s}).$$

Let the initial distribution of $\boldsymbol{z}_0$ be $p_0 \in \mathcal{P}(\mathcal{Z})$, and let each one-dimensional CTMC $(z^i(t))$ have a local generator $\boldsymbol{\lambda}_t(i, \boldsymbol{z}) \coloneqq [\lambda_t^{s \to s'}(i, \tilde{\boldsymbol{z}}^{i,s})]_{s,s' \in S}$, that is a mapping $\boldsymbol{\lambda} : [0, T] \times V \times \mathcal{Z} \to \mathbb{R}^{|S| \times |S|}$. Local transition rates are defined as

$$\lambda_t^{z^i \to s}(i, \boldsymbol{z}) = \begin{cases} \lim_{\Delta t \downarrow 0} p_{t+\Delta t|t}^i(s \mid \boldsymbol{z}), & s \neq z^i, \\ -\sum_{s' \neq z^i} \lambda_t^{z^i \to s'}(i, \boldsymbol{z}), & s = z^i. \end{cases}$$

As we are interested in working with non-homogeneous Markov chains, recovering the Markov kernels from the rate matrix is non-trivial and requires commutativity assumptions of the rate matrix [Norris, 1998]. For simplicity, we only consider arbitrarily small time intervals $0 < \Delta t \ll 1$ and adopt a "piece-wise" approximation to the rate matrix, such that it is constant for the interval $[t, t + \Delta t)$. A similar approximation is adopted in the context of generative modelling, by both discrete diffusion [Sun et al., 2022] and flow matching [Campbell et al., 2024]. We can then express each site-marginal Markov transition kernel as

$$q_{t+\Delta t|t}^i(s \mid \boldsymbol{z}) \approx \delta_{s, z^i} + \Delta t \lambda_t^{z^i \to s}(i, \boldsymbol{z}) + o(\Delta t), \quad i \in V. \tag{14}$$

The dynamics at each site $i \in V$ can be described by *full conditional* master equations, i.e. defined conditionally on a global configuration fixed at all sites but $i$. These correspond to

$$
\begin{aligned}
& \partial_t q_t^{i|-i}(z^i \mid \boldsymbol{z}) \\
&= \lim_{\Delta t \to 0} \Delta t^{-1} \left[ q_{t+\Delta t}^{i|-i}(z^i \mid \boldsymbol{z}) - q_t^{i|-i}(z^i \mid \boldsymbol{z}) \right] \\
&= \lim_{\Delta t \to 0} \Delta t^{-1} \left[ \sum_{s \in S} q_{t+\Delta t|t}^i(z^i \mid \tilde{\boldsymbol{z}}^{i,s}) q_t^{i|-i}(s \mid \boldsymbol{z}) - q_t^{i|-i}(z^i \mid \boldsymbol{z}) \right] \\
&= \lim_{\Delta t \to 0} \Delta t^{-1} \sum_{s \neq z^i} \left[ q_{t+\Delta t|t}^i(z^i \mid \tilde{\boldsymbol{z}}^{i,s}) q_t^{i|-i}(s \mid \boldsymbol{z}) - q_{t+\Delta t|t}^i(s \mid \boldsymbol{z}) q_t^{i|-i}(z^i \mid \boldsymbol{z}) \right] \\
&= \sum_{s \neq z^i} \left[ \lambda_t^{s \to z^i}(i, \tilde{\boldsymbol{z}}^{i,s}) q_t^{i|-i}(s \mid \boldsymbol{z}) - \lambda_t^{z^i \to s}(i, \boldsymbol{z}) q_t^{i|-i}(z^i \mid \boldsymbol{z}) \right] .
\end{aligned}
\tag{15}
$$

In matrix form, this can be written as $\partial_t q_t^{i|-i}(- \mid \boldsymbol{z}) = \boldsymbol{\lambda}_t(i, \boldsymbol{z})^\top q_t^{i|-i}(- \mid \boldsymbol{z})$ for the probability vector $q_t^{i|-i}(- \mid \boldsymbol{z}) \in \Delta^{|S|}$.

**Independent infinitesimal transitions.** Consider two global configurations $\tilde{\boldsymbol{z}}, \boldsymbol{z} \in \mathcal{Z}$, such that $\tilde{\boldsymbol{z}} \neq \boldsymbol{z}$. At time $t \in [0, 1]$ and for $0 < \Delta t \ll 1$, we assume independent transitions along each coordinate and adopt the approximation in (14), so that

$$
\begin{aligned}
q_{t+\Delta t|t}(\tilde{\boldsymbol{z}} \mid \boldsymbol{z}) &= \prod_{i \in V} q_{t+\Delta t|t}^i(\tilde{z}^i \mid \boldsymbol{z}) \\
&\approx \prod_{i \in V} \left[ \delta_{z^i, \tilde{z}^i} + \Delta t \lambda_t^{z^i \to \tilde{z}^i}(i, \boldsymbol{z}) + o(\Delta t) \right] \\
&= \delta_{\boldsymbol{z}, \tilde{\boldsymbol{z}}} + \Delta t \sum_{i \in V} \lambda_t^{z^i \to \tilde{z}^i}(i, \boldsymbol{z}) \prod_{j \neq i} \delta_{\tilde{z}^j, z^j} + o(\Delta t) .
\end{aligned}
\tag{16}
$$

Notice that the appropriateness of this assumption is highly dependent on the process we are modeling. It is probably a safe assumption for models of propagation on a graph, but it might not be for scenarios where sites are strongly coupled, such as object tracking. In this latter case, a site switching to a state of occupancy would imply that a neighboring site has switched to a state of inoccupancy at the exact same time, which couldn't be captured by the dependence structured described by (16).

**Global master equation.** We can characterize the generator of a CTMC $\boldsymbol{\Lambda}_t = [\Lambda_t(\tilde{\boldsymbol{z}} \mid \boldsymbol{z})]_{\tilde{\boldsymbol{z}}, \boldsymbol{z} \in \mathcal{Z}}$ by populating it with asynchronous site-wise transitions

$$
\Lambda_t(\tilde{\boldsymbol{z}} \mid \boldsymbol{z}) = \sum_{i \in V} \lambda_t^{z^i \to \tilde{z}^i}(i, \boldsymbol{z}) \prod_{j \neq i} \delta_{\tilde{z}^j, z^j}
\tag{17}
$$

and letting $\Lambda_t(\boldsymbol{z} \mid \boldsymbol{z}) = -\sum_{\tilde{\boldsymbol{z}} \neq \boldsymbol{z}} \Lambda_t(\tilde{\boldsymbol{z}} \mid \boldsymbol{z})$. In other words, the only non-zero entries of the generator are those representing transitions at a single site, and there are at most $|V| \times |S| \times |S|$ of those, as compared to the $|S|^{|V|} \times |S|^{|V|}$ entries of the matrix. The ME can then be expressed as

$$
\begin{aligned}
\partial_t q_t(\boldsymbol{z}) &= \sum_{\tilde{\boldsymbol{z}} \neq \boldsymbol{z}} [\Lambda_t(\boldsymbol{z} \mid \tilde{\boldsymbol{z}}) q_t(\tilde{\boldsymbol{z}}) - \Lambda_t(\tilde{\boldsymbol{z}} \mid \boldsymbol{z}) q_t(\boldsymbol{z})] \\
&= \sum_{i \in V} \sum_{s \neq z^i} \left[ \lambda_t^{s \to z^i}(i, \tilde{\boldsymbol{z}}^{i,s}) q_t^{i|-i}(s \mid \boldsymbol{z}) - \lambda_t^{z^i \to s}(i, \boldsymbol{z}) q_t^{i|-i}(z^i \mid \boldsymbol{z}) \right] q_t^{-i}(\boldsymbol{z}) \\
&= \sum_{i \in V} \partial_t q_t^{i|-i}(z^i \mid \boldsymbol{z}) q_t^{-i}(\boldsymbol{z}) .
\end{aligned}
\tag{18}
$$

$$
\tag{19}
$$

## C.4   Derivation of $D_{\mathrm{KL}}(Q \,\|\, P)$

Consider two CTMCs with path measures $Q, P \in \mathcal{P}(\Omega_{[0,T]})$, and denote their respective rate matrices entries with $\Lambda_t(\tilde{\boldsymbol{z}} \mid \boldsymbol{z})$ and $\Psi_t(\tilde{\boldsymbol{z}} \mid \boldsymbol{z})$ for $\boldsymbol{z}, \tilde{\boldsymbol{z}} \in \mathcal{Z}$. Their KL divergence, as discussed in Opper

and Sanguinetti [2007], Seifner and Sánchez [2023], can be derived from the limit of discrete-time transitions with step size $h := T/K$ as

$$D_{\mathrm{KL}}(Q||P)$$

$$= \lim_{K \to \infty} \sum_{\boldsymbol{z}_{0:K}} q_0(\boldsymbol{z}_0) \prod_{k=0}^{K-1} q_{k+h|k}(\boldsymbol{z}_{k+h} \mid \boldsymbol{z}(t_k)) \log \frac{q_0(\boldsymbol{z}_0) \prod_{k=0}^{K-1} q_{k+h|k}(\boldsymbol{z}_{k+h} \mid \boldsymbol{z}(t_k))}{p_0(\boldsymbol{z}_0) \prod_{k=0}^{K-1} p_{k+h|k}(\boldsymbol{z}_{k+h} \mid \boldsymbol{z}(t_k))}$$

$$= \sum_{\boldsymbol{z}_0} q_0(\boldsymbol{z}_0) \log \frac{q_0(\boldsymbol{z}_0)}{p_0(\boldsymbol{z}_0)} + \lim_{K \to \infty} \sum_{k=0}^{K-1} \mathbb{E}_{q_k(\boldsymbol{z})} \left[ \sum_{\boldsymbol{z}_{k+h}} q_{k+h|k}(\boldsymbol{z}_{k+h} \mid \boldsymbol{z}) \log \frac{q_{k+h|k}(\boldsymbol{z}_{k+h} \mid \boldsymbol{z})}{p_{k+h|k}(\boldsymbol{z}_{k+h} \mid \boldsymbol{z})} \right]$$

$$\tag{20}$$

$$= D_{\mathrm{KL}}(q_0||p_0) + \int_0^T \mathbb{E}_{q_t(\boldsymbol{z})} \sum_{\tilde{\boldsymbol{z}} \neq \boldsymbol{z}} \left\{ \Psi_t(\tilde{\boldsymbol{z}} \mid \boldsymbol{z}) + \Lambda_t(\tilde{\boldsymbol{z}} \mid \boldsymbol{z}) \left( \log \frac{\Lambda_t(\tilde{\boldsymbol{z}} \mid \boldsymbol{z})}{\Psi_t(\tilde{\boldsymbol{z}} \mid \boldsymbol{z})} - 1 \right) \right\} dt,$$

where the last line follows from dividing and multiplying each summand in (20) by $h$, and substituting the transition probabilities with rates,

$$\frac{q_{k+h|k}(\boldsymbol{z}_{k+h} \mid \boldsymbol{z})}{h} \log \frac{q_{k+h|k}(\boldsymbol{z}_{k+h} \mid \boldsymbol{z})}{p_{k+h|k}(\boldsymbol{z}_{k+h} \mid \boldsymbol{z})} \xrightarrow{h \to 0} \begin{cases} \Lambda_t(\boldsymbol{z}_{k+h} \mid \boldsymbol{z}) \log \frac{\Lambda_t(\boldsymbol{z}_{k+h}|\boldsymbol{z})}{\Psi_t(\boldsymbol{z}_{k+h}|\boldsymbol{z})} & \boldsymbol{z}_{k+h} \neq \boldsymbol{z}, \\ \sum_{\tilde{\boldsymbol{z}} \neq \boldsymbol{z}} [\Psi_t(\tilde{\boldsymbol{z}} \mid \boldsymbol{z}) - \Lambda_t(\tilde{\boldsymbol{z}} \mid \boldsymbol{z})] & \boldsymbol{z}_{k+h} = \boldsymbol{z}. \end{cases}$$

By assuming transition probabilities of the form $q_{t+h|t}(\tilde{\boldsymbol{z}} \mid \boldsymbol{z}) = \prod_{i \in V} q_{t+h|t}(\tilde{\boldsymbol{z}}(i) \mid \mathcal{N}_i(\boldsymbol{z}))$ where we define a neighborhood $\mathcal{N}_i(\boldsymbol{z}) := \{z^i, \boldsymbol{z}(j) : i \sim j\}$, we can rewrite each summand in (20) as

$$\mathbb{E}_{q_k(\boldsymbol{z})} \left[ \sum_{\boldsymbol{z}_{k+h}} q_{k+h|k}(\boldsymbol{z}_{k+h} \mid \boldsymbol{z}) \log \frac{q_{k+h|k}(\boldsymbol{z}_{k+h} \mid \boldsymbol{z})}{p_{k+h|k}(\boldsymbol{z}_{k+h} \mid \boldsymbol{z})} \right]$$

$$= \mathbb{E}_{q_k(\boldsymbol{z})} \left[ \sum_{i \in V} \sum_{s \in S} q_{k+h, k}^i(s \mid \mathcal{N}_i(\boldsymbol{z})) \log \frac{q_{k+h, k}^i(s \mid \mathcal{N}_i(\boldsymbol{z}))}{p_{k+h, k}^i(s \mid \mathcal{N}_i(\boldsymbol{z}))} \right].$$

Letting $K \to \infty$ and plugging (8), we get

$$D_{\mathrm{KL}}(Q||P)$$

$$= D_{\mathrm{KL}}(q_0||p_0) + \int_0^T \mathbb{E}_{q_t(\boldsymbol{z})} \sum_{i \in V} \sum_{s \neq z^i} \left\{ \psi_t^{z^i \to s}(i, \boldsymbol{z}) - \lambda_t^{z^i \to s}(i, \boldsymbol{z}) \right.$$

$$\left. + \lambda_t^{z^i \to s}(i, \boldsymbol{z}) \left( \log \frac{\lambda_t^{z^i \to s}(i, \boldsymbol{z})}{\psi_t^{z^i \to s}(i, \boldsymbol{z})} \right) \right\} dt. \tag{21}$$

### C.5 Derivation of the evidence lower bound

We start by proving a simple but fundamental property of the solution to equation 2, by showing that the optimal paths in latent space are Markovian, provided our reference process $P \in \mathcal{P}(\Omega_{[0,T]})$ is Markovian. This motivates our parameterization of such process as a CTMC.

**Lemma 3** ($Q^\star$ is Markov). *If $P \in \mathcal{P}(\Omega_{[0,T]})$ is Markov, then $Q^\star := \varphi_\# \mathbb{Q}^\star$ solving equation 2 with reference measure $\mathrm{P}((d\boldsymbol{z}(t))_{t \in [0,T]}, d\boldsymbol{x}_{1:K}) := \prod_{k \in [K]} p(d\boldsymbol{x}_k \mid \boldsymbol{z}(t_k)) P((d\boldsymbol{z}(t))_{t \in [0,T]})$ is Markov.*

*Proof.* The proof is a simple extension of Léonard [2013, Prop. 2.10] to the case where the process is latent, and we restate it here for completeness.

We consider an arbitrary time $t \in [0, T]$. When it is an observation time, i.e. $t = t_k$ for some $k = 1, \ldots, K$, we consider a fixed time-marginal at $t_k$ in observation space, denoted $\hat{q}_k \in \mathcal{P}(\mathcal{X})$, a conditional measure at $t_k$ in latent space $\hat{q}_{t_k}(\cdot|\boldsymbol{x}_k) \in \mathcal{P}(\mathcal{Z})$, and conditional path measures on both latent trajectories and observations, before and after $t$. These can be denoted

as $\hat{Q}^<_{\cdot|\boldsymbol{z}(t_k)} := \hat{Q}^{[0,t_k)}_{\cdot|\boldsymbol{z}(t_k)} \in \mathcal{P}(\Omega_{[0,t_k)} \times \mathcal{X}_{[0,t_k)})$ and $\hat{Q}^>_{\cdot|\boldsymbol{z}(t_k)} := \hat{Q}^{(t_k,T]}_{\cdot|\boldsymbol{z}(t_k)} \in \mathcal{P}(\Omega_{(t_k,T]} \times \mathcal{X}_{(t_k,T]})$, where we denote, with a slight abuse of notation, $\mathcal{X}_{[0,t_k)}$ and $\mathcal{X}_{(t_k,T]}$ to be the product space of observations happening before and after $t_k$. When $t$ is not an observation time, we simply consider a prescribed time-marginal in latent space $\hat{q}_t \in \mathcal{P}(\mathcal{Z})$ and the conditional path measures $\hat{Q}^<_{\cdot|\boldsymbol{z}(t)} := \hat{Q}^{[0,t)}_{\cdot|\boldsymbol{z}(t)} \in \mathcal{P}(\Omega_{[0,t)} \times \mathcal{X}_{[0,t)})$ and $\hat{Q}^>_{\cdot|\boldsymbol{z}(t)} := \hat{Q}^{(t,T]}_{\cdot|\boldsymbol{z}(t)} \in \mathcal{P}(\Omega_{(t_k,T]} \times \mathcal{X}_{(t_k,T]})$. As the proof in this case naturally follows from that of Léonard [2013, Prop. 2.10], we focus our attention to the case where $t$ is an observation time $t_k$.

We want to prove that, among all the joint measures Q that satisfy $q_k = \hat{q}_k$, $q_{t_k}(\cdot|\boldsymbol{x}_k) = \hat{q}_{t_k}(\cdot|\boldsymbol{x}_k)$, $Q^<_{\cdot|\boldsymbol{z}(t_k)} = \hat{Q}^<_{\cdot|\boldsymbol{z}(t_k)}$ and $Q^>_{\cdot|\boldsymbol{z}(t_k)} = \hat{Q}^>_{\cdot|\boldsymbol{z}(t_k)}$, a minimum in the KL divergence is attained by

$$\int_{\mathcal{X}} \int_{\mathcal{Z}} \hat{Q}^<_{\cdot|\boldsymbol{z}(t_k)} \otimes \hat{Q}^>_{\cdot|\boldsymbol{z}(t_k)} \hat{q}_{t_k}(d\boldsymbol{z}(t_k)|\boldsymbol{x}_k) \hat{q}_k(d\boldsymbol{x}_k), \tag{22}$$

i.e. the latent process is Markov [Léonard, 2013]. By arbitrariness of $t_k$ and of the measures we fix, this is also true for the solution to equation 2. This can be shown by applying the additive property of the KL divergence twice, conditioning on a $\boldsymbol{x}_k$ and $\boldsymbol{z}(t_k)$ first,

$$D_{\mathrm{KL}}(Q\,\|\,P) = D_{\mathrm{KL}}(\hat{q}_{t_k}(\cdot|\boldsymbol{x}_k)\hat{q}_k\,\|\,p_{t_k}(\cdot|\boldsymbol{x}_k)p_k) + \int_{\mathcal{X}} \int_{\mathcal{Z}} D_{\mathrm{KL}}(Q_{\cdot|\boldsymbol{z}_k}\,\|\,P_{\cdot|\boldsymbol{z}_k})\hat{q}_{t_k}(d\boldsymbol{z}(t_k)|\boldsymbol{x}_k)\hat{q}_k(d\boldsymbol{x}_k),$$

and then on the prescribed half path $\hat{Q}^<_{\cdot|\boldsymbol{z}(t_k)}$, obtaining

$$D_{\mathrm{KL}}(Q_{\cdot|\boldsymbol{z}_k}\,\|\,P_{\cdot|\boldsymbol{z}_k}) = D_{\mathrm{KL}}(\hat{Q}^<_{\cdot|\boldsymbol{z}(t_k)}\,\|\,P^<_{\cdot|\boldsymbol{z}(t_k)}) + \int_{\Omega_{(t_k,T]}} D_{\mathrm{KL}}\left(Q^{[t_k,T]}_{\cdot|(\boldsymbol{z}(t))_{t\in[0,t_k]}}\,\|\,P^>_{\cdot|\boldsymbol{z}(t_k)}\right) d\hat{Q}^<_{\cdot|\boldsymbol{z}(t_k)}.$$

By Jensen's inequality, we get

$$D_{\mathrm{KL}}(Q^>_{\cdot|\boldsymbol{z}(t_k)}\,\|\,P^>_{\cdot|\boldsymbol{z}(t_k)}) = D_{\mathrm{KL}}\left(\int_{\Omega_{(t_k,T]}} Q^{[t_k,T]}_{\cdot|(\boldsymbol{z}(t))_{t\in[0,t_k]}} d\hat{Q}^<_{\cdot|\boldsymbol{z}(t_k)}\,\middle\|\,P^>_{\cdot|\boldsymbol{z}(t_k)}\right)$$
$$\leq \int_{\Omega_{(t_k,T]}} D_{\mathrm{KL}}\left(Q^{[t_k,T]}_{\cdot|(\boldsymbol{z}(t))_{t\in[0,t_k]}}\,\|\,P^>_{\cdot|\boldsymbol{z}(t_k)}\right) d\hat{Q}^<_{\cdot|\boldsymbol{z}(t_k)},$$

and equality is achieved if and only if the process is Markov, i.e. $Q^{[t_k,T]}_{\cdot|(\boldsymbol{z}(t))_{t\in[0,t_k]}} = \hat{Q}^>_{\cdot|\boldsymbol{z}(t_k)}$. This proves that a minimum satisfying the prescribed marginals is achieved by a Markov process, i.e. satisfying equation 22. $\qquad\square$

Next, we derive the evidence lower bound for noiseless data, as presented in Proposition 2, and for noisy data. Alternative derivations for the latter can be found in Opper and Sanguinetti [2007], Wildner and Koeppl [2019].

Our derivation follows by analyzing the limit of discretized processes, following an approach analogous to the derivation of the KL divergence between two CTMCs in Opper and Sanguinetti [2007]. Specifically, we consider probability mass functions corresponding to marginal and conditionals of a discretized CTMC $(\boldsymbol{z}(t))_{t\in[t_1,t_K]}$ on a uniform grid $t_k = \tau_k^0 < \tau_k^1 < \cdots < \tau_k^{T_k-1} < \tau_k^{T_k} = t_{k+1}$, where $T_k = (t_{k+1} - t_k)/\Delta t$, for $k = 1, \ldots, K-1$. We then let $\Delta t \to 0$, and simultaneously $T_k \to \infty$. We denote the latent process at a discrete time $\tau$ as $\boldsymbol{z}_\tau := \boldsymbol{z}(\tau)$.

### C.5.1 Noiseless data - Proof of Proposition 2

When $\boldsymbol{x}_{1:K}$ is a noiseless observation of a Markov process $(\boldsymbol{z}(t))_{t\in[t_1,t_K]}$ at times $t_{1:K}$, we can leverage the Markov property and obtain at any time $\tau_k^j$, for $j=1,\ldots,T_k-2$ and $k=1,\ldots,K-1$

$$
\begin{aligned}
\bar{p}_{\tau_k^j}(\boldsymbol{z}) &:= p_{\tau_k^j}(\boldsymbol{z}\,|\,\boldsymbol{x}_{1:K}) \\
&= p_{\tau_k^j}(\boldsymbol{z}\,|\,\boldsymbol{x}_k,\boldsymbol{x}_{k+1}) \\
&= p_{\tau_k^j\,|\,t_k}(\boldsymbol{z}\,|\,\boldsymbol{x}_k)\frac{p_{t_{k+1}\,|\,\tau_k^j}(\boldsymbol{x}_{k+1}\,|\,\boldsymbol{z})}{p_{t_{k+1}\,|\,t_k}(\boldsymbol{x}_{k+1}\,|\,\boldsymbol{x}_k)}, \\
\bar{p}_{\tau_k^{j+1}\,|\,\tau_k^j}(\tilde{\boldsymbol{z}}\,|\,\boldsymbol{z}) &:= p_{\tau_k^{j+1}\,|\,\tau_k^j}(\tilde{\boldsymbol{z}}\,|\,\boldsymbol{z},\boldsymbol{x}_{1:K}) \\
&= p_{\tau_k^{j+1}\,|\,\tau_k^j}(\tilde{\boldsymbol{z}}\,|\,\boldsymbol{z},\boldsymbol{x}_{k+1}) \\
&= p_{\tau_k^{j+1}\,|\,\tau_k^j}(\tilde{\boldsymbol{z}}\,|\,\boldsymbol{z})\frac{p_{t_{k+1}\,|\,\tau_k^{j+1}}(\boldsymbol{x}_{k+1}\,|\,\tilde{\boldsymbol{z}})}{p_{t_{k+1}\,|\,\tau_k^j}(\boldsymbol{x}_{k+1}\,|\,\boldsymbol{z})}.
\end{aligned}
$$

Hence,

$$
\begin{aligned}
p(\boldsymbol{z}_{\tau_k^1:\tau_k^{T_k-1}}\,|\,\boldsymbol{x}_{1:K}) &= \bar{p}_{\tau_k^1}(\boldsymbol{z}_{\tau_k^1})\prod_{j=1}^{T_k-2}\bar{p}_{\tau_k^{j+1}\,|\,\tau_k^j}(\boldsymbol{z}_{\tau_k^{j+1}}\,|\,\boldsymbol{z}_{\tau_k^j}) \\
&= p_{\tau_k^1\,|\,t_k}(\boldsymbol{z}_{\tau_k^1}\,|\,\boldsymbol{x}_k)\prod_{j=1}^{T_k-2}p_{\tau_k^{j+1}\,|\,\tau_k^j}(\boldsymbol{z}_{\tau_k^{j+1}}\,|\,\boldsymbol{z}_{\tau_k^j})\frac{p_{t_{k+1}\,|\,\tau_k^{T_k-1}}(\boldsymbol{x}_{k+1}\,|\,\boldsymbol{z}_{\tau_k^{T_k-1}})}{p_{t_{k+1}\,|\,t_k}(\boldsymbol{x}_{k+1}\,|\,\boldsymbol{x}_k)}.
\end{aligned}
$$

It follows that

$$
\begin{aligned}
&D_{\mathrm{KL}}\left(Q_{\cdot|\boldsymbol{x}_{1:K}}^\theta\,||\,P_{\cdot|\boldsymbol{x}_{1:K}}\right) \\
&= \sum_{k=1}^{K-1}\mathbb{E}_{q_{\tau_k^1:\tau_k^{T_k-1}}^\theta(\cdot\,|\,\boldsymbol{x}_k,\boldsymbol{x}_{k+1})}\left[\log\frac{q_{\tau_k^1:\tau_k^{T_k-1}}^\theta(\boldsymbol{z}_{\tau_k^1:\tau_k^{T_k-1}}\,|\,\boldsymbol{x}_k,\boldsymbol{x}_{k+1})}{\bar{p}_{\tau_k^1}(\boldsymbol{z}_{\tau_k^1})\prod_{j=1}^{T_k-2}\bar{p}_{\tau_k^{j+1}\,|\,\tau_k^j}(\boldsymbol{z}_{\tau_k^{j+1}}\,|\,\boldsymbol{z}_{\tau_k^j})}\right] \\
&= \sum_{k=1}^{K-1}\mathbb{E}_{q_{\tau_k^1:\tau_k^{T_k-1}}^\theta(\cdot\,|\,\boldsymbol{x}_k,\boldsymbol{x}_{k+1})}\left[\log\frac{q_{\tau_k^1:\tau_k^{T_k-1}}^\theta(\boldsymbol{z}_{\tau_k^1:\tau_k^{T_k-1}}\,|\,\boldsymbol{x}_k,\boldsymbol{x}_{k+1})}{p_{\tau_k^1\,|\,t_k}(\boldsymbol{z}_{\tau_k^1}\,|\,\boldsymbol{x}_k)\prod_{j=1}^{T_k-2}p_{\tau_k^{j+1}\,|\,\tau_k^j}(\boldsymbol{z}_{\tau_k^{j+1}}\,|\,\boldsymbol{z}_{\tau_k^j})}\right] \\
&\quad - \mathbb{E}_{q_{\tau_k^{T_k-1}}^\theta(\cdot\,|\,\boldsymbol{x}_k,\boldsymbol{x}_{k+1})}\left[\log p_{t_{k+1}\,|\,\tau_k^{T_k-1}}(\boldsymbol{x}_{k+1}\,|\,\boldsymbol{z}_{\tau_k^{T_k-1}})\right] + \log p_{t_{k+1}\,|\,t_k}(\boldsymbol{x}_{k+1}\,|\,\boldsymbol{x}_k)
\end{aligned}
$$

Denoting $\log Z = \sum_{k=1}^{K-1}\log p_{t_{k+1}\,|\,t_k}(\boldsymbol{x}_{k+1}\,|\,\boldsymbol{x}_k)$ and as $\Delta t\to 0$ and $T_k\to\infty$, we get

$$
= \log Z + \sum_{k=1}^{K-1}D_{\mathrm{KL}}\left(Q_{\cdot|\boldsymbol{x}_k,\boldsymbol{x}_{k+1}}^\theta\,||\,P_{\cdot|\boldsymbol{x}_k}\right) - \mathbb{E}_{q_{t_{k+1}^-}^\theta(\cdot|\boldsymbol{x}_k,\boldsymbol{x}_{k+1})}\left[\log p_{t_{k+1}\,|\,t_{k+1}^-}(\boldsymbol{x}_{t_{k+1}}\,|\,\boldsymbol{z}(t_{k+1}^-))\right],
$$

where each KL term is restricted to the time interval $(t_k,t_{k+1})$ and $\boldsymbol{z}(t_{k+1}^-)=\lim_{t\to t_k^-}\boldsymbol{z}(t)$.

### C.5.2 Noisy data

When $\boldsymbol{x}_{1:K}$ is a noisy observation of $(\boldsymbol{z}(t))_{t\in[t_1,t_K]}$ at times $t_{1:K}$, at any time $\tau_k^j$ for $j=1,\ldots,T_k-1$ and $k=1,\ldots,K-1$,

$$
\bar{p}_{\tau_k^j}(\boldsymbol{z}_{\tau_k^j}) := p_{\tau_k^j}(\boldsymbol{z}_{\tau_k^j}\,|\,\boldsymbol{x}_{1:K}) \tag{23}
$$

$$
= p_{\tau_k^j}(\boldsymbol{z}_{\tau_k^j})\frac{p_{\leq\tau_k^j|\tau_k^j}(\boldsymbol{x}_{\leq\tau_k^j}\,|\,\boldsymbol{z}_{\tau_k^j})p_{>\tau_k^j|\tau_k^j}(\boldsymbol{x}_{>\tau_k^j}\,|\,\boldsymbol{z}_{\tau_k^j})}{p_{1:K}(\boldsymbol{x}_{1:K})}, \tag{24}
$$

$$
\bar{p}_{\tau_k^{j+1}|\tau_k^j}(\boldsymbol{z}_{\tau_k^{j+1}}\,|\,\boldsymbol{z}_{\tau_k^j}) := p_{\tau_k^{j+1}|\tau_k^j}(\boldsymbol{z}_{\tau_k^{j+1}}\,|\,\boldsymbol{z}_{\tau_k^j},\boldsymbol{x}_{1:K}) \tag{25}
$$

$$
= p_{\tau_k^{j+1}|\tau_k^j}(\boldsymbol{z}_{\tau_k^{j+1}}\,|\,\boldsymbol{z}_{\tau_k^j})\frac{p_{>\tau_k^j|\tau_k^{j+1}}(\boldsymbol{x}_{>\tau_k^j}\,|\,\boldsymbol{z}_{\tau_k^{j+1}})}{p_{>\tau_k^j|\tau_k^j}(\boldsymbol{x}_{>\tau_k^j}\,|\,\boldsymbol{z}_{\tau_k^j})}. \tag{26}
$$

At the last step before an observation time, we can further decompose

$$p_{>\tau_k^{T_k-1} | \tau_k^{T_k}}(\boldsymbol{x}_{>\tau_k^{T_k-1}} \mid \boldsymbol{z}_{\tau_k^{T_k}}) = p_{\geq t_{k+1} | t_{k+1}}(\boldsymbol{x}_{\geq t_{k+1}} \mid \boldsymbol{z}_{t_{k+1}})$$

$$= p_{t_{k+1}}(\boldsymbol{x}_{k+1} \mid \boldsymbol{z}_{t_{k+1}}) p_{>t_{k+1} | t_{k+1}}(\boldsymbol{x}_{>t_{k+1}} \mid \boldsymbol{z}_{t_{k+1}}).$$

Hence, denoting $t_1 : t_k = \{t_1, \tau_1^1, \ldots, \tau_{K-1}^{T_{K-1}-1}, t_K\}$, we get

$$p_{t_1:t_k}(\boldsymbol{z}_{t_1:t_k} \mid \boldsymbol{x}_{1:K}) = \bar{p}_{t_1}(\boldsymbol{z}_{t_1}) \prod_{k=1}^{K-1} \prod_{j=0}^{T_k-1} \bar{p}_{\tau_k^{j+1} | \tau_k^j}(\boldsymbol{z}_{\tau_k^{j+1}} \mid \boldsymbol{z}_{\tau_k^j})$$

$$= \frac{p_{t_1}(\boldsymbol{z}_{t_1}) p_{t_1}(\boldsymbol{x}_1 \mid \boldsymbol{z}_{t_1})}{p_{1:K}(\boldsymbol{x}_{1:K})} \prod_{k=1}^{K-1} p_{t_{k+1}}(\boldsymbol{x}_{k+1} \mid \boldsymbol{z}_{t_{k+1}}) \prod_{j=0}^{T_k-1} p_{\tau_k^{j+1} | \tau_k^j}(\boldsymbol{z}_{\tau_k^{j+1}} \mid \boldsymbol{z}_{\tau_k^j}).$$

and

$$q_{t_1:t_K}^\theta(\boldsymbol{z}_{t_1:t_K} \mid \boldsymbol{x}_{1:K}) = q_{t_1}^\theta(\boldsymbol{z}_{t_1} \mid \boldsymbol{x}_{1:K}) \prod_{k=1}^{K-1} q_{\tau_k^1:t_{k+1}}^\theta(\boldsymbol{z}_{\tau_k^1:t_{k+1}} \mid \boldsymbol{z}_{t_k}, \boldsymbol{x}_{>t_k}).$$

Denoting $\log Z = \log p_{1:K}(\boldsymbol{x}_{1:K})$, the KL can be expressed as

$$D_{\mathrm{KL}}\left(Q_{\cdot|\boldsymbol{x}_{1:K}}^\theta \,||\, P_{\cdot|\boldsymbol{x}_{1:K}}\right)$$

$$= \mathbb{E}_{q_{t_1:t_K}^\theta(\cdot \mid \boldsymbol{x}_{1:K})}\left[\log \frac{q_{t_1:t_K}^\theta(\boldsymbol{z}_{t_1:t_K} \mid \boldsymbol{x}_{1:K})}{p_{t_1:t_k}(\boldsymbol{z}_{t_1:t_k} \mid \boldsymbol{x}_{1:K})}\right]$$

$$= \log Z + \mathbb{E}_{q_{t_1:t_K}^\theta(\cdot \mid \boldsymbol{x}_{1:K})}\left[\log \frac{q_{t_1:t_K}^\theta(\boldsymbol{z}_{t_1:t_K} \mid \boldsymbol{x}_{1:K})}{p_{t_1:t_k}(\boldsymbol{z}_{t_1:t_k})}\right] - \mathbb{E}_{q_{t_1:t_K}^\theta(\cdot \mid \boldsymbol{x}_{1:K})}\left[\sum_{k=1}^K \log p_{t_k}(\boldsymbol{x}_k \mid \boldsymbol{z}_{t_k})\right].$$

As $\Delta t \to 0$ and $T_k \to \infty$, we get

$$= \log Z + D_{\mathrm{KL}}(Q_{\cdot|\boldsymbol{x}_{1:K}}^\theta \,||\, P) - \mathbb{E}_{Q_{\cdot|\boldsymbol{x}_{1:K}}^\theta}\left[\sum_{k=1}^K \log p_{t_k}(\boldsymbol{x}_k \mid \boldsymbol{z}_{t_k})\right]$$

### C.6  Unconditional loss

In this section, we aim at justifying the choice of the surrogate loss in Section 3.2. We do so by bounding its distance to the ideal loss, with respect to the Markov process $Q^\star \in \mathcal{P}(\Omega_{[t_1,t_K]})$ that is unavailable.

**Definition 4.** *For a given time t, we define:*

- *The total variation distance:*

$$\left\|q_t^\star - q_t^\theta\right\|_{TV} = \mathbb{E}_{\pi_{1:K}(\boldsymbol{x}_{1:K})}\left[\sum_{\boldsymbol{z} \in \mathcal{Z}} \left|q_t^\star(\boldsymbol{z} \mid \boldsymbol{x}_{1:K}) - q_t^\theta(\boldsymbol{z} \mid \boldsymbol{x}_{1:K})\right|\right],$$

- *The expected Lambda difference:*

$$\varepsilon_t^\Lambda(\theta) := \mathbb{E}_{q_t^\theta(\boldsymbol{z}, \boldsymbol{x}_{>t})} \sum_{\tilde{\boldsymbol{z}} \neq \boldsymbol{z}} \left|\Lambda_t^\star(\tilde{\boldsymbol{z}} | \boldsymbol{z}, \boldsymbol{x}_{>t}) - \Lambda_t^\theta(\tilde{\boldsymbol{z}} | \boldsymbol{z}, \boldsymbol{x}_{>t})\right|.$$

**Theorem 5.** *The following bound holds:*

$$\left|\mathcal{L}_{KL}(\phi) - \hat{\mathcal{L}}_{KL}^\theta(\phi)\right| \leq \int_{t_1}^{t_K} \left\|q_t^\star - q_t^\theta\right\|_{TV} \cdot A_t(\theta, \phi) \, dt,$$

*where*

$$A_t(\theta, \phi) = \mathbb{E}_{q_t^\theta(\boldsymbol{z}, \boldsymbol{x}_{1:K})}\left[\varepsilon_t^\Lambda(\theta) \max_{\tilde{\boldsymbol{z}} \neq \boldsymbol{z}} \left|\log \Lambda_t^\phi(\tilde{\boldsymbol{z}} | \boldsymbol{z})\right| - \Lambda_t^\phi(\boldsymbol{z} | \boldsymbol{z})\right].$$

*Proof.*

**Lemma 6.**

$$D_{\mathrm{KL}}(Q^\star \,|\, Q^\phi) \propto \mathbb{E}_{\pi_{1:K}}\left[ D_{\mathrm{KL}}(Q^\star_{\cdot|\boldsymbol{x}_{1:K}} \,|\, Q^\phi)\right]. \tag{27}$$

*Proof.* Let $Q^\phi$ have rates $\Lambda_t^\phi(-|-)$, $Q^\star$ have rates $\Lambda_t^\star(-|-)$, and $Q^\star_{\cdot|\boldsymbol{x}_{1:K}}$ have rates $\Lambda_t^\star(-|-,\boldsymbol{x}_{>t})$, where we use the shorthand $\boldsymbol{x}_{>t} := \{\boldsymbol{x}_k : t_k > t,\ k \in 1,\dots,K\}$. Then,

$$\mathbb{E}_{\pi_{1:K}}\left[ D_{\mathrm{KL}}(Q^\star_{\cdot|\boldsymbol{x}_{1:K}} \,|\, Q^\phi)\right]$$

$$\propto \mathbb{E}_{\pi_{1:K}}\left[ \int_{t_1}^{t_K} \mathbb{E}_{q_t^\star(\boldsymbol{z}|\boldsymbol{x}_{1:K})} \sum_{\tilde{\boldsymbol{z}} \neq \boldsymbol{z}} \left\{ \Lambda_t^\phi(\tilde{\boldsymbol{z}} \mid \boldsymbol{z}) - \Lambda_t^\star(\tilde{\boldsymbol{z}} \mid \boldsymbol{z}, \boldsymbol{x}_{>t}) \log \Lambda_t^\phi(\tilde{\boldsymbol{z}} \mid \boldsymbol{z}) \right\} dt \right]$$

$$= \mathbb{E}_{\pi_{1:K}}\left[ \int_{t_1}^{t_K} \mathbb{E}_{q_t^\star(\boldsymbol{z}|\boldsymbol{x}_{1:K})} \sum_{\tilde{\boldsymbol{z}} \neq \boldsymbol{z}} \left\{ \Lambda_t^\phi(\tilde{\boldsymbol{z}} \mid \boldsymbol{z}) - \Lambda_t^\star(\tilde{\boldsymbol{z}} \mid \boldsymbol{z}) \frac{q_{>t|t}^\star(\boldsymbol{x}_{>t} \mid \tilde{\boldsymbol{z}})}{q_{>t|t}^\star(\boldsymbol{x}_{>t} \mid \boldsymbol{z})} \log \Lambda_t^\phi(\tilde{\boldsymbol{z}} \mid \boldsymbol{z}) \right\} dt \right]$$

$$= \int_{t_1}^{t_K} \mathbb{E}_{\pi_{1:K}} \mathbb{E}_{q_t^\star(\boldsymbol{z}|\boldsymbol{x}_{1:K})} \sum_{\tilde{\boldsymbol{z}} \neq \boldsymbol{z}} \left\{ \Lambda_t^\phi(\tilde{\boldsymbol{z}} \mid \boldsymbol{z}) - \Lambda_t^\star(\tilde{\boldsymbol{z}} \mid \boldsymbol{z}) \frac{q_{>t|t}^\star(\boldsymbol{x}_{>t} \mid \tilde{\boldsymbol{z}})}{q_{>t|t}^\star(\boldsymbol{x}_{>t} \mid \boldsymbol{z})} \log \Lambda_t^\phi(\tilde{\boldsymbol{z}} \mid \boldsymbol{z}) \right\} dt.$$

Applying Fubini's theorem for interchanging integrals,

$$= \int_{t_1}^{t_K} \mathbb{E}_{q_t^\star(\boldsymbol{z})} \mathbb{E}_{q_{1:K|t}^\star(\boldsymbol{x}_{1:K}|\boldsymbol{z})} \sum_{\tilde{\boldsymbol{z}} \neq \boldsymbol{z}} \left\{ \Lambda_t^\phi(\tilde{\boldsymbol{z}} \mid \boldsymbol{z}) - \Lambda_t^\star(\tilde{\boldsymbol{z}} \mid \boldsymbol{z}) \frac{q_{>t|t}^\star(\boldsymbol{x}_{>t} \mid \tilde{\boldsymbol{z}})}{q_{>t|t}^\star(\boldsymbol{x}_{>t} \mid \boldsymbol{z})} \log \Lambda_t^\phi(\tilde{\boldsymbol{z}} \mid \boldsymbol{z}) \right\} dt$$

$$= \int_{t_1}^{t_K} \mathbb{E}_{q_t^\star(\boldsymbol{z})} \sum_{\tilde{\boldsymbol{z}} \neq \boldsymbol{z}} \left\{ \Lambda_t^\phi(\tilde{\boldsymbol{z}} \mid \boldsymbol{z}) - \mathbb{E}_{q_{1:K|t}^\star(\boldsymbol{x}_{1:K}|\boldsymbol{z})} \left[ \frac{q_{>t|t}^\star(\boldsymbol{x}_{>t} \mid \tilde{\boldsymbol{z}})}{q_{>t|t}^\star(\boldsymbol{x}_{>t} \mid \boldsymbol{z})} \right] \Lambda_t^\star(\tilde{\boldsymbol{z}} \mid \boldsymbol{z}) \log \Lambda_t^\phi(\tilde{\boldsymbol{z}} \mid \boldsymbol{z}) \right\} dt$$

$$= \int_{t_1}^{t_K} \mathbb{E}_{q_t^\star(\boldsymbol{z})} \sum_{\tilde{\boldsymbol{z}} \neq \boldsymbol{z}} \left\{ \Lambda_t^\phi(\tilde{\boldsymbol{z}} \mid \boldsymbol{z}) - \Lambda_t^\star(\tilde{\boldsymbol{z}} \mid \boldsymbol{z}) \log \Lambda_t^\phi(\tilde{\boldsymbol{z}} \mid \boldsymbol{z}) \right\} dt$$

$$\propto D_{\mathrm{KL}}(Q^\star \,||\, Q^\phi).$$

$\square$

However, we do not have access to $\Lambda^\star(-|-,\boldsymbol{x}_{>t})$ and $q_t^\star(-|\boldsymbol{x}_{1:K})$, but to their approximations $\Lambda^\theta(-|-,\boldsymbol{x}_{>t})$ and $q_t^\theta(-|\boldsymbol{x}_{1:K})$. Let $q_t^\star(\boldsymbol{z}, \boldsymbol{x}_{1:K}) := q_t^\star(\boldsymbol{z}|\boldsymbol{x}_{1:K})\pi(\boldsymbol{x}_{1:K})$. For simplicity, we break down each KL term into parts, so to get

$$\mathbb{E}_{\pi_{1:K}}\left[ D_{\mathrm{KL}}(Q^\star_{\cdot|\boldsymbol{x}_{1:K}} \,|\, Q^\phi)\right] = \int_{t_1}^{t_K} \underbrace{\mathbb{E}_{q_t^\star(\boldsymbol{z},\boldsymbol{x}_{1:K})} \sum_{\tilde{\boldsymbol{z}} \neq \boldsymbol{z}} \Lambda_t^\phi(\tilde{\boldsymbol{z}} \mid \boldsymbol{z})}_{L_t^{(1)}(\boldsymbol{x}_{1:K})} + \underbrace{\mathbb{E}_{q_t^\star(\boldsymbol{z},\boldsymbol{x}_{1:K})} \sum_{\tilde{\boldsymbol{z}} \neq \boldsymbol{z}} \Lambda_t^\star(\tilde{\boldsymbol{z}} \mid \boldsymbol{z}, \boldsymbol{x}_{>t})}_{L_t^{(2)}(\boldsymbol{x}_{1:K})}$$

$$\underbrace{- \mathbb{E}_{q_t^\star(\boldsymbol{z},\boldsymbol{x}_{1:K})} \sum_{\tilde{\boldsymbol{z}} \neq \boldsymbol{z}} \Lambda_t^\star(\tilde{\boldsymbol{z}} \mid \boldsymbol{z}, \boldsymbol{x}_{>t}) \log \Lambda_t^\phi(\tilde{\boldsymbol{z}} \mid \boldsymbol{z})}_{-L_t^{(3)}(\boldsymbol{x}_{1:K})} dt,$$

$$\mathbb{E}_{\pi_{1:K}}\left[ D_{\mathrm{KL}}(Q^\theta_{\cdot|\boldsymbol{x}_{1:K}} \,|\, Q^\phi)\right] = \int_{t_1}^{t_K} \underbrace{\mathbb{E}_{q_t^\theta(\boldsymbol{z},\boldsymbol{x}_{1:K})} \sum_{\tilde{\boldsymbol{z}} \neq \boldsymbol{z}} \Lambda_t^\phi(\tilde{\boldsymbol{z}} \mid \boldsymbol{z})}_{\hat{L}_t^{(1)}(\boldsymbol{x}_{1:K})} + \underbrace{\mathbb{E}_{q_t^\theta(\boldsymbol{z},\boldsymbol{x}_{1:K})} \sum_{\tilde{\boldsymbol{z}} \neq \boldsymbol{z}} \Lambda_t^\theta(\tilde{\boldsymbol{z}} \mid \boldsymbol{z}, \boldsymbol{x}_{>t})}_{\hat{L}_t^{(2)}(\boldsymbol{x}_{1:K})}$$

$$\underbrace{- \mathbb{E}_{q_t^\theta(\boldsymbol{z},\boldsymbol{x}_{1:K})} \sum_{\tilde{\boldsymbol{z}} \neq \boldsymbol{z}} \Lambda_t^\theta(\tilde{\boldsymbol{z}} \mid \boldsymbol{z}, \boldsymbol{x}_{>t}) \log \Lambda_t^\phi(\tilde{\boldsymbol{z}} \mid \boldsymbol{z})}_{-\hat{L}_t^{(3)}(\boldsymbol{x}_{1:K})} dt.$$

Finally, we let

$$\mathbb{E}_{\pi_{1:K}}\left[ D_{\mathrm{KL}}(Q^\star_{\cdot|\boldsymbol{x}_{1:K}} \,|\, Q^\phi) - D_{\mathrm{KL}}(Q^\theta_{\cdot|\boldsymbol{x}_{1:K}} \,|\, Q^\phi)\right] = \int_{t_1}^{t_K} \sum_{i=1}^{3} \underbrace{L_t^{(i)}(\boldsymbol{x}_{1:K}) - \hat{L}_t^{(i)}(\boldsymbol{x}_{1:K})}_{D^{(i)}(\boldsymbol{x}_{1:K})} dt.$$

We quantify the error in terms of total variation distance and expected absolute error of the generator at each time $t \in [t_1, t_K]$,

$$\left\| q_t^\star - q_t^\theta \right\|_{TV} := \mathbb{E}_{\pi_{1:K}(\boldsymbol{x}_{1:K})} \left[ \sum_{\boldsymbol{z} \in \mathcal{Z}} \left| q_t^\star(\boldsymbol{z} \mid \boldsymbol{x}_{1:K}) - q_t^\theta(\boldsymbol{z} \mid \boldsymbol{x}_{1:K}) \right| \right]$$

$$= \mathbb{E}_{q_t^\theta(\boldsymbol{z}, \boldsymbol{x}_{1:K})} \left| \frac{q_t^\star(\boldsymbol{z}, \boldsymbol{x}_{1:K})}{q_t^\theta(\boldsymbol{z}, \boldsymbol{x}_{1:K})} - 1 \right|,$$

$$\varepsilon_t^\Lambda(\theta) := \mathbb{E}_{q_t^\theta(\boldsymbol{z}, \boldsymbol{x}_{>t})} \sum_{\tilde{\boldsymbol{z}} \neq \boldsymbol{z}} \left| \Lambda_t^\star(\tilde{\boldsymbol{z}} | \boldsymbol{z}, \boldsymbol{x}_{>t}) - \Lambda_t^\theta(\tilde{\boldsymbol{z}} | \boldsymbol{z}, \boldsymbol{x}_{>t}) \right|.$$

Then, we are interested in isolating the terms in $\left| \mathcal{L}_{\mathrm{KL}}(\phi) - \hat{\mathcal{L}}_{\mathrm{KL}}^\theta(\phi) \right|$ that depend on $\phi$,

$$\left| \mathbb{E}_{\pi_{1:K}} \left[ D_{\mathrm{KL}}(Q^\star_{\cdot | \boldsymbol{x}_{1:K}} \mid Q^\phi) - D_{\mathrm{KL}}(Q^\theta_{\cdot | \boldsymbol{x}_{1:K}} \mid Q^\phi) \right] \right| \leq \int_{t_1}^{t_K} \left| D^{(1)}(\boldsymbol{x}_{1:K}) \right| + \left| D^{(3)}(\boldsymbol{x}_{1:K}) \right| dt.$$

By applying Jensen's inequality and the Cauchy-Schwarz inequality, we can further bound these quantities as

$$\left| D^{(1)}(\boldsymbol{x}_{1:K}) \right| = \left| \mathbb{E}_{q_t^\theta(\boldsymbol{z}, \boldsymbol{x}_{1:K})} \left[ \left( \frac{q_t^\star(\boldsymbol{z}, \boldsymbol{x}_{1:K})}{q_t^\theta(\boldsymbol{z}, \boldsymbol{x}_{1:K})} - 1 \right) \sum_{\tilde{\boldsymbol{z}} \neq \boldsymbol{z}} \Lambda_t^\phi(\tilde{\boldsymbol{z}} \mid \boldsymbol{z}) \right] \right|$$

$$\leq \left\| q_t^\star - q_t^\theta \right\|_{TV} \mathbb{E}_{q_t^\theta(\boldsymbol{z}, \boldsymbol{x}_{1:K})} \left[ -\Lambda_t^\phi(\boldsymbol{z} \mid \boldsymbol{z}) \right],$$

$$\left| D^{(3)}(\boldsymbol{x}_{1:K}) \right|$$

$$= \left| \mathbb{E}_{q_t^\theta(\boldsymbol{z}, \boldsymbol{x}_{1:K})} \left[ \left( 1 - \frac{q_t^\star(\boldsymbol{z}, \boldsymbol{x}_{1:K})}{q_t^\theta(\boldsymbol{z}, \boldsymbol{x}_{1:K})} \right) \sum_{\tilde{\boldsymbol{z}} \neq \boldsymbol{z}} \left( \Lambda_t^\star(\tilde{\boldsymbol{z}} | \boldsymbol{z}, \boldsymbol{x}_{>t}) - \Lambda_t^\theta(\tilde{\boldsymbol{z}} | \boldsymbol{z}, \boldsymbol{x}_{>t}) \right) \log \Lambda_t^\phi(\tilde{\boldsymbol{z}} \mid \boldsymbol{z}) \right] \right|$$

$$\leq \left\| q_t^\star - q_t^\theta \right\|_{TV} \mathbb{E}_{q_t^\theta(\boldsymbol{z}, \boldsymbol{x}_{1:K})} \left| \sum_{\tilde{\boldsymbol{z}} \neq \boldsymbol{z}} \left( \Lambda_t^\star(\tilde{\boldsymbol{z}} | \boldsymbol{z}, \boldsymbol{x}_{>t}) - \Lambda_t^\theta(\tilde{\boldsymbol{z}} | \boldsymbol{z}, \boldsymbol{x}_{>t}) \right) \log \Lambda_t^\phi(\tilde{\boldsymbol{z}} \mid \boldsymbol{z}) \right|$$

$$\leq \varepsilon_t^\Lambda(\theta) \left\| q_t^\star - q_t^\theta \right\|_{TV} \mathbb{E}_{q_t^\theta(\boldsymbol{z}, \boldsymbol{x}_{1:K})} \left[ \max_{\tilde{\boldsymbol{z}} \neq \boldsymbol{z}} \left| \log \Lambda_t^\phi(\tilde{\boldsymbol{z}} \mid \boldsymbol{z}) \right| \right].$$

Hence,

$$\left| \mathcal{L}_{\mathrm{KL}}(\phi) - \hat{\mathcal{L}}_{\mathrm{KL}}^\theta(\phi) \right|$$

$$\leq \int_{t_1}^{t_K} \left\| q_t^\star - q_t^\theta \right\|_{TV} \left( \varepsilon_t^\Lambda(\theta) \mathbb{E}_{q_t^\theta(\boldsymbol{z}, \boldsymbol{x}_{1:K})} \left[ \max_{\tilde{\boldsymbol{z}} \neq \boldsymbol{z}} \left| \log \Lambda_t^\phi(\tilde{\boldsymbol{z}} \mid \boldsymbol{z}) \right| \right] - \mathbb{E}_{q_t^\theta(\boldsymbol{z}, \boldsymbol{x}_{1:K})} \left[ \Lambda_t^\phi(\boldsymbol{z} \mid \boldsymbol{z}) \right] \right) dt$$

$$\square$$

# D  Implementation details

## D.1  Architecture

**Self-Omitted Attention**  Given a configuration $\boldsymbol{z} \in \mathcal{Z}$, observation and next observation times $t, t_{\mathrm{next}} \in \mathbb{R}$, a representation of future observations $\boldsymbol{x}_{\mathrm{next}}$, and context $\boldsymbol{c}$, we parameterize conditional local generators of the form $(t, t_{\mathrm{next}}, \boldsymbol{z}, \boldsymbol{x}_{\mathrm{next}}, \boldsymbol{c}) \mapsto \boldsymbol{\Lambda}_{t, t_{\mathrm{next}}}(\boldsymbol{z}, \boldsymbol{x}_{\mathrm{next}}, \boldsymbol{c}) \in \mathbb{R}^{|V| \times |S| \times |S|}$. We denote the output at a specific site $i \in V$ as $\lambda_{t, t_{\mathrm{next}}}^{s \to \tilde{s}, \theta}(i, \boldsymbol{z}, \boldsymbol{x}_{\mathrm{next}}, \boldsymbol{c})$. For a given hidden dimension $d$, we use multi-layer perceptrons to compute site-wise representations $\boldsymbol{e}^i = f\left(x_{\mathrm{next}}^i, c^i\right) \in \mathbb{R}^d$ and $\tilde{\boldsymbol{e}}^i = f\left(z^i, x_{\mathrm{next}}^i, c^i\right) \in \mathbb{R}^d$, that we collect in matrices $\mathbf{E}, \tilde{\mathbf{E}} \in \mathbb{R}^{|V| \times d}$. The unconditional setting reflects that of the conditional model, but without the $t_{\mathrm{next}}$ and $\boldsymbol{x}_{\mathrm{next}}$ terms. We group the columns of each matrix into $H$ attention heads $\mathbf{E}_1, \ldots, \mathbf{E}_H$ and $\tilde{\mathbf{E}}_1, \ldots, \tilde{\mathbf{E}}_H$ (such that $d \mod H = 0$), and

denote the representations of site $i$ in head $h$ as $e_h^i$, $\tilde{e}_h^i$. Moreover, we let $\tau = h(t, t_{\text{next}})$ be a time embedding.

We modify the attention mechanism so that the output at each site $i \in V$ is invariant to the input state $z^i$ at that site. This naturally follows from the fact that we are trying to parameterize transition rates for each site from any given (local) state to any other, while capturing neighborhood interactions. We do so by considering the usual query-key weight matrices $\mathbf{W}_{\mathbf{Q}_h}, \mathbf{W}_{\mathbf{K}_h} \in \mathbb{R}^{d \times d/H}$, the value matrix $\mathbf{W}_{\mathbf{V}} \in \mathbb{R}^{d \times d}$, and an additional matrix $\mathbf{W}_{\tilde{\mathbf{K}}_h} \in \mathbb{R}^{d \times d/H}$. We denote the site-specific queries and keys as $q_h^i = e_h^i \mathbf{W}_{\mathbf{Q}_h}$, $k_h^i = e_h^i \mathbf{W}_{\mathbf{K}_h}$ in $\mathbb{R}^{d/H}$, and an additional term $\tilde{k}_h^i = \tilde{e}_h^i \mathbf{W}_{\tilde{\mathbf{K}}_h} \in \mathbb{R}^{d/H}$ that includes state information, for $i \in V$ and $h = 1, \ldots, H$. We then compute the matrix $\mathbf{A}_h \in \mathbb{R}^{|V| \times |V|}$ by letting each element be

$$a_h^{ij} = \text{softmax}\left(\left\{\hat{a}_h^{il}/\sqrt{d/H},\ l \in V\right\}\right), \quad \hat{a}_h^{ij} = \begin{cases} \langle q_h^i, \tilde{k}_h^j \rangle, & i \sim j, \\ \langle q_h^i, k_h^j \rangle, & i = j, \\ 0, & \text{otherwise.} \end{cases}$$

When the neighborhood structure is that of a lattice (and denoting $M = |\mathcal{N}_i|$ for any $i$), we use the method proposed in the Vision Transformer Cellular Automata [Tesfaldet et al., 2022] to localize attention, reducing computations from $\mathcal{O}(|V|^2)$ to $\mathcal{O}(|V|M)$. For graphs with an arbitrary neighborhood structure, we perform element-wise masking of $\mathbf{A}$ with the adjacency matrix.

Considering the values $\mathbf{V}_h = \mathbf{E}_h \mathbf{W}_{\mathbf{V}_h} \in \mathbb{R}^{|V| \times d/H}$, the self-omitted attention output $\text{SOA}_h \in \mathbb{R}^{|V| \times d/H}$ is then computed and information across heads is combined by concatenating them, as

$$\mathbf{O} = \text{concat}\left[\text{SOA}_1, \ldots, \text{SOA}_H\right] \in \mathbb{R}^{|V| \times d}, \quad \text{SOA}_h = \mathbf{A}_h \mathbf{V}_h \in \mathbb{R}^{|V| \times d/H}.$$

The off-diagonal elements of the rate matrix for each site are then computed by passing each $o^i$ through an MLP mapping to $\mathbb{R}^{|S| \times (|S|-1)}$. Filling the diagonals with the row-wise sum and concatenating the matrices yields the local generator in $\mathbb{R}^{|V| \times |S| \times |S|}$.

## D.2 Training

The simulation algorithms that can be used at training time for trajectory reconstruction are reported in Algorithm 1 and Algorithm 2. Notice that it is also possible to learn the unconditional generator at the same time as the unconditional one, by freezing the gradients of $\theta$ before updating the $\hat{\mathcal{L}}_{\text{KL}}^\theta$ loss. While all datapoints in a batch are processed in parallel, we might need to evolve the solver through different time points for each batch. This is feasible by applying the tricks for parallel solving of neural ODEs with varying time-intervals presented in Chen et al. [2021].

While training the conditional generator, we often observed the model converge to a local minima where the next observed state is reached in a very short time right after the previous observation, and rates are then zeroed until the next observation time. This biases the distribution of samples seen at training time by the unconditional model, that might then experience "mode collapse" and predict all of the transition rates to be zero. This reflects the insight given by Theorem 5. We found that choosing priors that bias the conditional model towards performing fewer transitions helps addressing this issue, as they tend to regularize the path.

## D.3 Computational considerations

Our method is not simulation-free, in the sense that learning is made possible by backpropagating through a solver. In doing so, a practitioner can incur in two fundamental problems, inaccurate gradients and memory-intensive training steps. The choice of a backpropagation technique can trade off one disadvantage for the other. In our experiments we use continuous adjoint methods, that provide memory-efficient numerical solutions (constant w.r.t. the time discretization grid) at the cost of incurring numerical errors that accumulate into potentially inaccurate gradient estimates. An overview of other possible approaches is presented in [Kidger, 2021].

---
**Algorithm 1** Forward simulation, conditional
---
**Require:** Observations $\boldsymbol{x}_1, \ldots, \boldsymbol{x}_K$ at times $t_1, \ldots, t_K$, step size $\Delta t$, future encoder $h_t$, initial encoder $q_1$, conditional local generator $\boldsymbol{\Lambda}_t$, prior $p_0$, $P$. *Optional:* context $\boldsymbol{c}_1, \ldots, \boldsymbol{c}_K$.
**Ensure:** Latent states $\boldsymbol{z}(t_1), \ldots, \boldsymbol{z}(t_K)$, KL of the path
1: Sample $\boldsymbol{z}(t_1) \sim q_1(\cdot | \boldsymbol{x}_1, h_{t_1}(\boldsymbol{x}_{>t_1}), \boldsymbol{c})$
2: $t_{\text{last}} \leftarrow t_1$
3: $\text{KL} \leftarrow D_{\text{KL}}(q_1 \,\|\, p_0)$
4: **for** $t \in (t_1, t_K]$ **do**
5:     Sample $\boldsymbol{z}$ from $q_{t+\Delta t | t}$, approximating equation 1 using $\boldsymbol{\Lambda}_t(\boldsymbol{z} \mid h_t(\boldsymbol{x}_{>t}), \boldsymbol{c}_{t_{\text{last}}})$
6:     Compute contribution $d\text{KL}_t$ to equation 21 at time $t$, using $\boldsymbol{\Lambda}_t(\boldsymbol{z} \mid h_t(\boldsymbol{x}_{>t}), \boldsymbol{c}_{t_{\text{last}}})$
7:     $\text{KL} \leftarrow \text{KL} + d\text{KL}_t \Delta t$
8:     **if** $t = t_k$ for $k = 1, \ldots, K$ **then**
9:         $\boldsymbol{z}(t_k) \leftarrow \boldsymbol{z}$
10:        $t_{\text{last}} \leftarrow t$
11:     **end if**
12: **end for**
---

---
**Algorithm 2** Neural master equation
---
**Require:** Observations $\boldsymbol{x}_1, \ldots, \boldsymbol{x}_K$ at times $t_1, \ldots, t_K$, step size $\Delta t$, future encoder $h_t$, initial encoder $q_1$, conditional local generator $\boldsymbol{\Lambda}_t$, prior $p_0$, $P$. *Optional:* context $\boldsymbol{c}_1, \ldots, \boldsymbol{c}_K$.
**Ensure:** Latent states $\boldsymbol{z}(t_1), \ldots, \boldsymbol{z}(t_K)$, KL of the path
1: Sample $\boldsymbol{z}(t_1) \sim q_1(\cdot | \boldsymbol{x}_1, h_{t_1}(\boldsymbol{x}_{>t_1}), \boldsymbol{c})$
2: $t_{\text{last}} \leftarrow t_1$
3: $\text{KL} \leftarrow D_{\text{KL}}(q_1 \,\|\, p_0)$
4: **for** $t \in (t_1, t_K]$ **do**
5:     Sample $\boldsymbol{z} \sim q_t = \prod_i q_t^i$ using the Gumbell-Softmax trick
6:     Compute $\frac{d}{dt} q_t^i$ for all $i \in V$, using $\boldsymbol{\Lambda}_t(\boldsymbol{z} \mid h_t(\boldsymbol{x}_{>t}), \boldsymbol{c}_{t_{\text{last}}})$ and $q_t^i$
7:     Compute contribution $d\text{KL}_t$ to equation 21 at time $t$, using $\boldsymbol{\Lambda}_t(\boldsymbol{z} \mid h_t(\boldsymbol{x}_{>t}), \boldsymbol{c}_{t_{\text{last}}})$
8:     $\text{KL} \leftarrow \text{KL} + d\text{KL}_t \Delta t$
9:     **if** $t = t_k$ for $k = 1, \ldots, K$ **then**
10:        $\boldsymbol{z}(t_k) \leftarrow \boldsymbol{z}$
11:        $t_{\text{last}} \leftarrow t$
12:     **end if**
13: **end for**
---

## E   Experiments

### E.1   Datasets

**Epidemics** The dataset is comprised of a collection of 250 random graphs with 128 nodes each and a given expected degree of 3, where edges are generated at random. Two covariates $\boldsymbol{c}_1^i$, $\boldsymbol{c}_2^i$ are generated for each node $i \in V$ by sampling from a standard normal distribution. An epidemic is then spread according to a Susceptible-Infected-Recovered (SIR) model [Keeling and Eames, 2005, Paré et al., 2020, Dolgov and Savostyanov, 2024]. Initially, all nodes are set to be susceptible ($S$) with the exception of $p_0$ nodes set to be infected ($I$) at random. Each graph in the dataset is evolved in the continuous-time interval $[0, 19]$, where a time-homogeneous functional form for the local transition

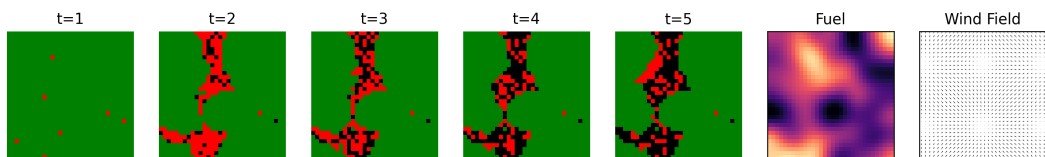

Figure 2: First 5 observations in time of a sequence from the wildfires dataset, with the corresponding covariates.



Figure 3: First 5 observations in time of a sequence from the epidemics dataset.

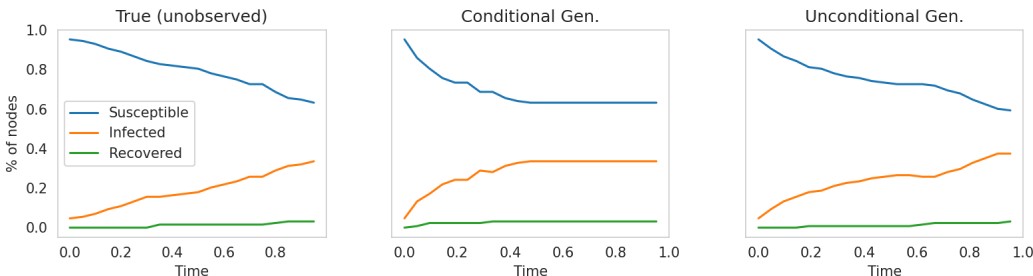

Figure 4: True and generated SIR curves in a time interval observed only at the two endpoints, in an held-out graph of 128 nodes.

rates from $S$ to $I$ and from $I$ to recovered ($R$) is specified as

$$\lambda^{S \to I}(i, \boldsymbol{x}) = \beta \exp\left(\sin(\mathbf{c}_1^i) + \cos(\mathbf{c}_2^i)\right) \left|\mathcal{N}_i^I\right|,$$
$$\lambda^{I \to R}(i, \boldsymbol{x}) = \gamma,$$

where $\mathcal{N}_i^I := \{j \in V \mid \boldsymbol{x}(j) = I, j \sim i\}$, $\beta = 6$ and $\gamma = 0.2$. These parameters do not correspond to physically meaningful quantities, and adjusting them to reflect real-world spread dynamics remains an interesting avenue for future work. Each graph is observed at $K = 20$ regularly spaced time points, with no observation noise (i.e., $\mathcal{X} \equiv \mathcal{Y}$). The data is simulated using $\tau-$leaping [Gillespie, 2001], with $\tau = 1 \times 10^{-2}$. A sample observed in its first 5 time steps is displayed in Figure 3.

**Wildfires** We consider 250 observations of $32^2-$dimensional lattice-valued data represented as images, where each pixel can take three possible values: unburned ($U$), burning ($B$), or extinguished ($E$). Spatially structured covariates corresponding to wind fields $\mathbf{w}$ and ground-level fuel $\mathbf{f}$ are generated at the same resolution. At time zero, each pixel is set to $B$ with a probability $p_0^B = 0.005$ (i.e., we expect 5 pixels to be burning), while all the others are set to $U$. The dynamic is then evolved in the continuous-time interval $[0, 19]$ by local transition rates with time-homogeneous functional forms

$$\lambda^{U \to B}(i, \boldsymbol{x}) = \text{ReLU}(a_0 + a_1 \mathbf{f}^i) \times \text{ReLU}\left(b_0 + b_1 \sum_{j \in \mathcal{N}_i^B} \mathbf{a}^{ij}\right),$$

$$\lambda^{E \to B}(i, \boldsymbol{x}) = \text{ReLU}(c_0 + c_1 \mathbf{f}^i) \times \text{ReLU}\left(d_0 + d_1 \sum_{j \in \mathcal{N}_i^B} \mathbf{a}^{ij}\right),$$

$$\lambda^{B \to E}(i, \boldsymbol{x}) = \gamma,$$

where $\mathcal{N}_i^B := \{j \in V \mid \boldsymbol{x}(j) = I, j \sim i\}$, and $\mathbf{a}^{ij}$ is a *wind alignment* value obtained by the dot product between the relative position of the neighbor $j$ w.r.t. $i$ and the value of the wind field at $j$. For our simulation, we set $a_0 = b_0 = c_0 = d_0 = 0.1$, $a_1 = 5$, $b_1 = 1$, $c_1 = d_1 = 0.01$, and $\gamma = 0.5$. Similarly to the first setting, each wildfire is observed at $K = 20$ regularly spaced time points with no observation noise. A sample observed in its first 5 time steps, as well as the related covariates, is displayed in Figure 2.

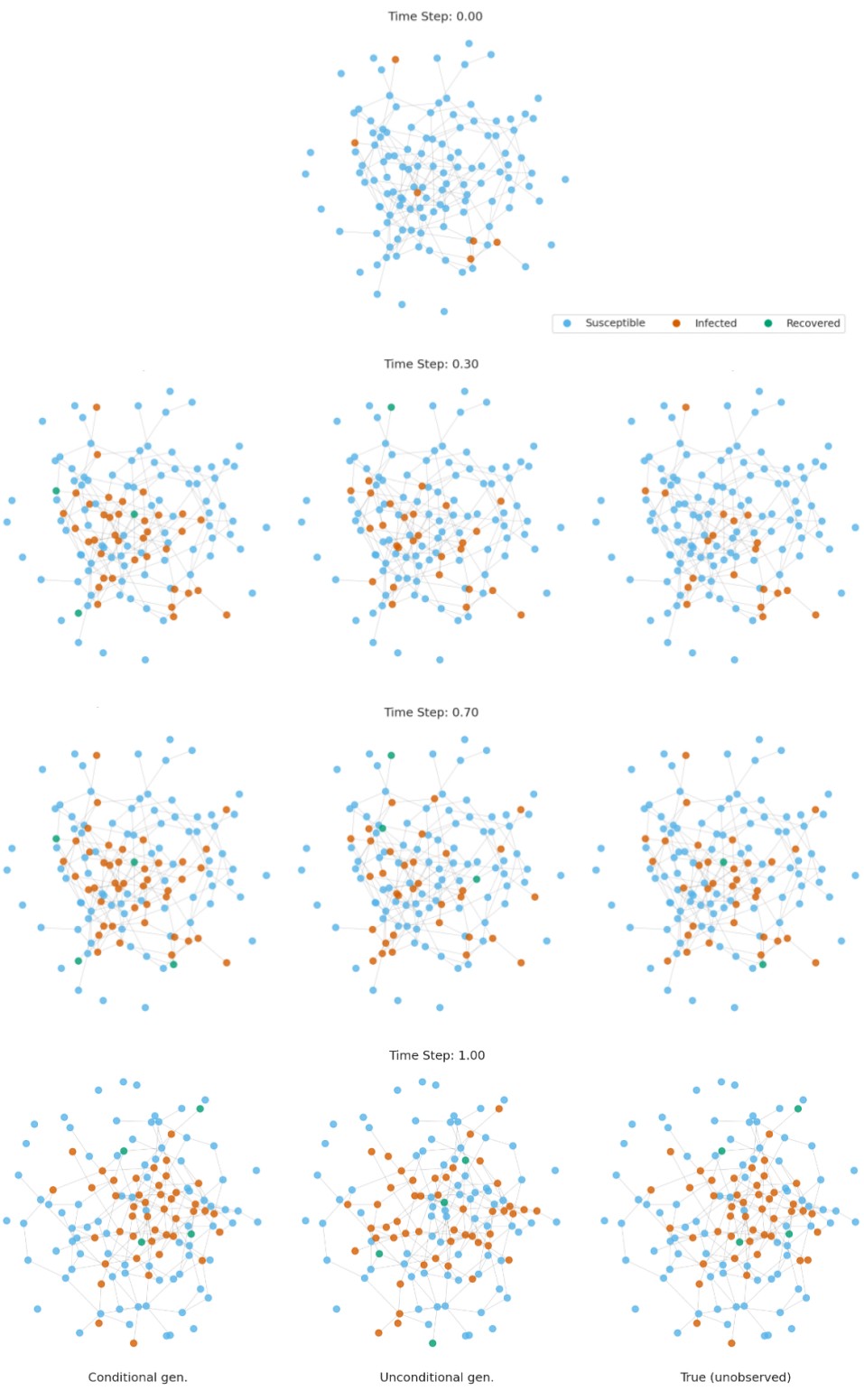

Figure 5: Evolution of an epidemic on an held-out graph. Endpoint-conditioned generation (left), unconditional generation (center), trajectory observed only at the endpoints (right).

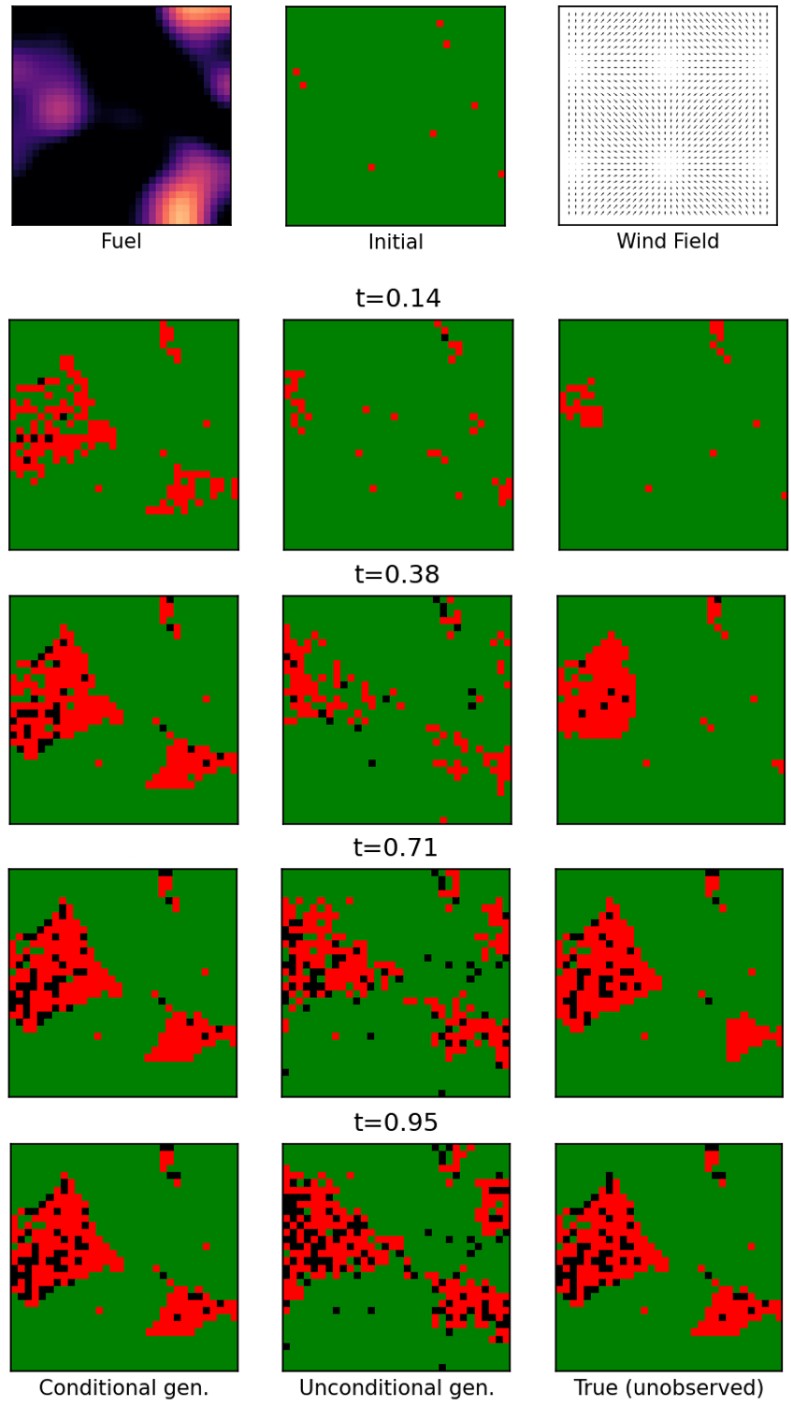

Figure 6: Initial conditions (top) and generated trajectories from the conditional (left) and unconditional (center) models, and true sequence observed only at the endpoints(right). Results shown for an held-out example.

### E.2 Model

Since there is no observation noise, all we need to parameterize in our experiments are the conditional and unconditional generators. Both can be thought of as mappings $\mathcal{X} \to \mathbb{R}_{\geq 0}^{|V| \times |S| \times |S|}$, i.e. the output shall be a local transition rate matrix at each site $i \in V$. For the wildfires experiment we simply consider a $3 \times 3$ Moore neighborhood, whereas for the epidemics we mask the attention matrix with the adjacency matrix of each observation. We constrain the output to be positive by applying a softmax function. We specify the prior path measure by a prior rate matrix, where we set to zero physically impossible transitions (e.g. $U \to E$ for wildfires, or $S \to R$ for epidemics) and the remaining off-diagonal elements to a constant value $c$. More complex functional forms are possible, and shall be chosen for example by simulating from the prior predictive distribution [Gelman et al., 2020].

### E.3 Results

We provide a qualitative overview of the results we have obtained so far. These shall be considered preliminary, and a quantitative comparison with other baselines (e.g. the mean-field approximation from Seifner and Sánchez [2023]) will be carried out in future work. For the epidemics dataset, we display generated trajectories on an held-out graph in Figure 5, as well as the aggregated SIR curves for the same example in Figure 4. Notice how the conditional model tends to converge quickly to the end solution, while the unconditional model mirrors the true unobserved trajectory more closely. For the wildfires experiments, we display results on held-out examples in Figure 6 and Figure 7. Despite the lack of information at the initial time, the unconditional model can still predict an evolution very close to the ground truth final configuration.

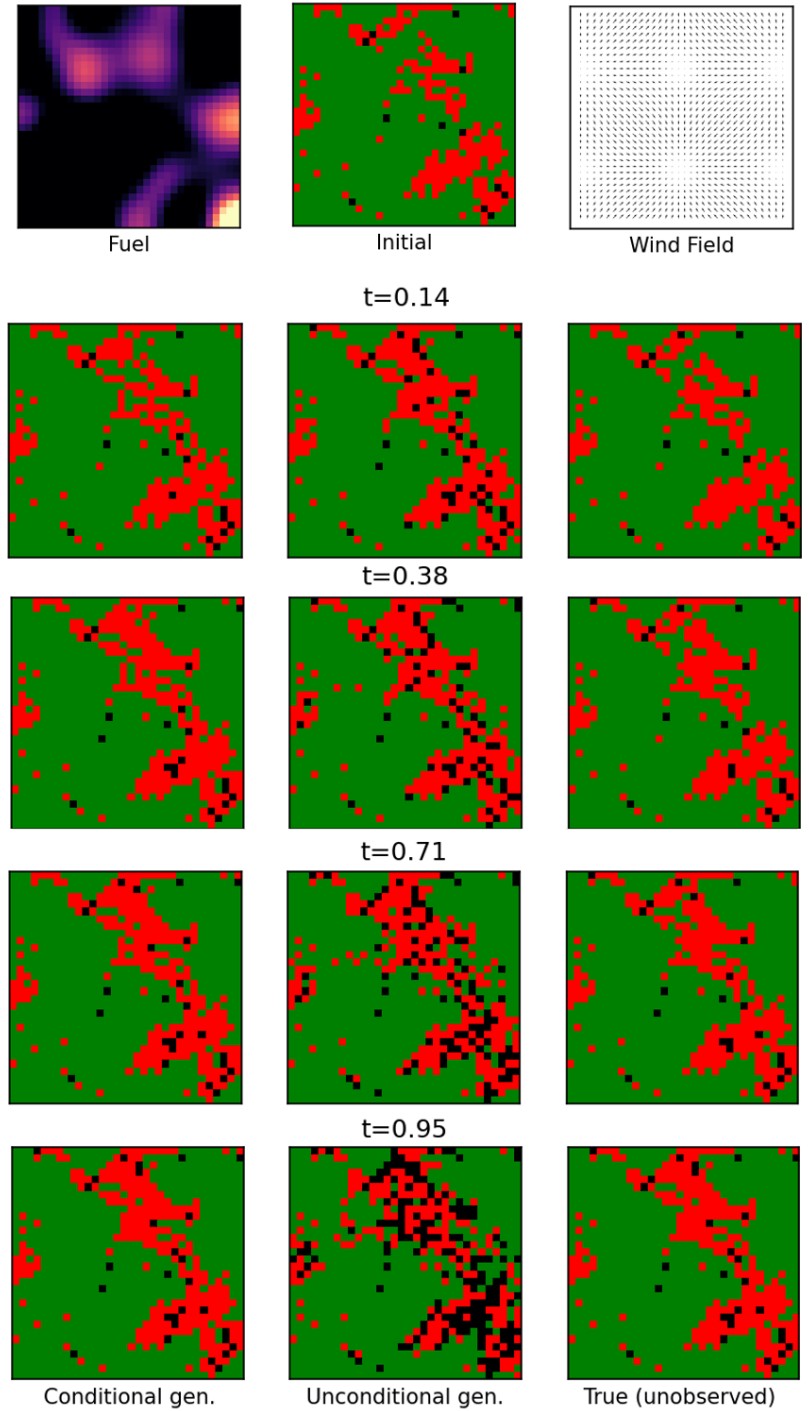

Figure 7: Same as Figure 6 but at a different stage of the simulated wildfire propagation, results shown for an held-out example.

