# OpenReview forum: "Variational Inference for Interacting Particle Systems with Discrete Latent States"
_NeurIPS.cc/2024/Workshop/BDU — NeurIPS BDU Workshop 2024 Poster_

### Official Review · Reviewer_FZ1z · 2024-10-07
**Need more details**

**Rating:** 6
**Confidence:** 5

**Review:**

Pro:

1. The paper introduces a novel Bayesian learning framework for interacting particle systems with discrete latent states, which addresses the challenge of inferring dynamics from partial, noisy observations. This is a valuable contribution to the field.
2. The paper formulates the problem as a multi-marginal Schrödinger bridge with aligned samples, which is a well-established and effective approach for solving similar problems.
3. The proposed method incorporates an emission distribution for decoding latent states, which is a practical and important component in real-world applications.

Cons:

1. Can you discuss the computational complexity and scalability of the proposed approach? What are the limitations and potential challenges in applying the method to high-dimensional spatio-temporal processes?
2. Can you provide more insights into the variational approximation used in the approach? What assumptions are made and how do these assumptions affect the accuracy of the inference?
3. Can you provide more information on the learning procedure? How did you optimize the evidence lower bound? Did you use any specific optimization algorithms or techniques?
4. How does the computational complexity of your approach scale with the dimensionality of the spatio-temporal processes? Have you performed any experiments to evaluate the scalability of your method?
5. How does the proposed method compare to existing methods for inferring dynamics from partial, noisy observations? Can you provide a more detailed discussion of the advantages and limitations of the proposed approach compared to previous approaches?

---

### Official Review · Reviewer_4L1V · 2024-10-07
**The paper titled "Variational Inference for Interacting Particle Systems with Discrete Latent States" proposes a novel Bayesian learning framework for modeling systems of interacting particles, with applications such as epidemic spread and wildfire propagation. The methodology is built around continuous-time Markov chains (CTMC) and employs a variational posterior to infer the latent dynamics from partial, noisy observations. This work introduces an efficient two-stage learning approach, leveraging the multi-marginal Schrödinger bridge, to reconstruct and predict trajectories. The paper’s advantages include the scalability of its variational approximation and the flexibility to integrate domain-specific knowledge through informative priors, which broadens its applicability to high-dimensional spatiotemporal systems.  However, the paper presents a few limitations. While the method shows potential in simulated scenarios, it remains to be validated on real-world data. The reliance on neural networks for parameterizing the CTMC generator adds computational complexity, which might pose challenges in practical applications with large-scale systems. Furthermore, while the use of neural cellular automata improves flexibility, the approach might require extensive hyperparameter tuning. When compared to existing literature, such as the use of mean-field variational approaches or classical cellular automata models, this work is more sophisticated but could benefit from more comparisons with other contemporary techniques to better establish its credibility and performance in diverse settings.  This paper aligns well with recent developments in variational inference for CTMCs and interacting particle systems, but further empirical testing and real-world applications are necessary to fully understand its potential impact.**

**Rating:** 7
**Confidence:** 4

**Review:**

----

Evaluation:

1. Quality:



The quality of this paper is creditable in terms of its methodological rigor and the clear articulation of a complex topic like variational inference for interacting particle systems (IPS). The authors successfully combine ideas from CTMCs, the Schrödinger bridge problem, and Bayesian learning frameworks to introduce a novel approach for inferring the dynamics of high-dimensional, spatiotemporal systems. The mathematical formulations are thorough, and the two-stage learning procedure is well-motivated and explained. While the paper primarily focuses on simulated results, the approach has the potential for real-world application once tested on empirical data. The theoretical development is solid, though the experimental validation could be more extensive.

2. Clarity:



The clarity of the paper is relatively strong in the theoretical sections, where the authors explain key concepts like variational inference and multi-marginal Schrödinger bridges. However, the presentation of the technical details might be dense for readers unfamiliar with the topic, which could limit accessibility to a broader audience. While the equations and algorithms are clearly structured, some sections could benefit from additional explanations or diagrams, particularly when discussing the architecture of neural cellular automata or the assumptions underlying the variational approximation. Additionally, clearer motivation for the choice of methods (e.g., why neural networks were chosen for parameterization) could enhance the reader's understanding.

3. Originality:



This paper presents a favorably original contribution by merging ideas from variational inference with interacting particle systems in the context of discrete latent states. The application of the multi-marginal Schrödinger bridge to this problem setting is innovative and offers a fresh perspective on trajectory inference and prediction. While interacting particle systems and Markov processes are well-established areas of study, this paper advances the field by introducing a novel, scalable inference approach that can be applied to high-dimensional data, which distinguishes it from traditional mean-field approximations or classical cellular automata methods. The originality of using neural networks in this context is another noteworthy aspect.

4.  Significance:



The significance of this work lies in its potential applicability to various real-world phenomena that can be modeled as interacting particle systems, such as epidemic modeling and wildfire prediction. The paper makes a valuable contribution by addressing the challenge of learning from partial and noisy observations in complex systems. If further validated through experiments on real-world datasets, this methodology could have far-reaching implications in fields ranging from epidemiology to environmental science and multi-agent systems. However, without real-world validation, the immediate significance of the results remains speculative, though the theoretical contributions are undoubtedly significant in advancing inference techniques for continuous-time stochastic systems.


List of Pros and Cons:


Pros:


- Novel Approach: Introduces an innovative two-stage learning procedure using the multi-marginal Schrödinger bridge for trajectory reconstruction and prediction.
- Scalability: The variational approximation is scalable, making it potentially suitable for high-dimensional spatiotemporal processes.
- Flexibility: The method allows for the incorporation of domain knowledge through informative priors, making it adaptable to different applications.
- Application Potential: The framework can be applied to diverse phenomena, such as epidemic spread and wildfire dynamics, offering  broad applicability.

Cons:



- Lack of Real-World Validation: The method has only been tested on simulated datasets, and its performance on real-world data remains to be demonstrated.
- Complexity: The use of neural networks for parameterizing the CTMC generator increases computational complexity, which might limit the method's scalability in very large systems.
- Accessibility: Some sections are mathematically dense, which might limit accessibility for non-expert readers or those not intimately familiar with the underlying concepts.
- Limited Experimental Evaluation: While the theoretical contributions are substantial, the paper could benefit from more extensive empirical evaluation, including comparisons with state-of-the-art methods.

Conclusion:


Overall, this paper is a high-quality, original contribution that proposes a novel approach for variational inference in interacting particle systems. While the paper is mathematically dense, it presents a significant advancement in the field of spatiotemporal modeling, particularly for systems governed by discrete latent states. The lack of real-world validation and the computational complexity introduced by the neural networks are drawbacks, but the paper's theoretical contributions and potential applications make it a valuable addition to the literature.

----

---

### Official Review · Reviewer_PYCF · 2024-10-08
**Comprehensive evaluation of "Variational Inference for Systems of Interacting Particles with Discrete Latent States"**

**Rating:** 7
**Confidence:** 4

**Review:**

This paper introduces a novel Bayesian learning framework for interacting particle systems (IPS) characterized by discrete latent states, with a special focus on learning from partial, noisy observations. The method innovatively exploits a variational posterior path metric to parameterize the generator of continuous-time Markov chains (CTMCs) to explain complex spatiotemporal processes.

---

### Official Review · Reviewer_e9yj · 2024-10-09
**Great theories and experiments, lack of bridge between theories and reality**

**Rating:** 6
**Confidence:** 5

**Review:**

The paper presents a Bayesian learning framework for interacting particle systems (IPS) with discrete latent states, addressing the challenge of inferring dynamics from partial and noisy observations. The authors introduce a scalable variational approximation and apply it to two simulated scenarios: epidemic trajectory inference on networks and wildfire spread prediction on lattices. The approach's strengths lie in its use of continuous-time Markov chains (CTMCs) and neural network parameterizations to effectively capture the underlying dynamic processes.

Improvement the authors can make:
1. Impact of Hyperparameters on Evolution Patterns
Now the hyperparameters on evolution patterns are fixed. For example:
- The epidemic model assumes a fixed random graph structure, where each node has an average of 3 neighbors.
-  β = 6 and γ = 0.2 etc.
This limits the generalization of the results to real-world scenarios, where social networks can be more heterogeneous. Exploring a broader range of network topologies would strengthen the model's applicability and allow for more generalized conclusions. Therefore, the paper would benefit from a deeper exploration of how key hyperparameters, such as infection probability, number of neighbors in the epidemic model, and wind speed or fuel availability in the wildfire model, influence the system's temporal and spatial dynamics. Analyzing these factors could reveal tipping points or significant shifts in behavior, providing valuable insights into the sensitivity of the models to these parameters.

2. More explanations for the hyper-parameters
As the paper mentions "These parameters do not correspond to physically meaningful quantities, and adjusting them to reflect real-world spread dynamics remains an interesting avenue for future work" in line 528-530, it would be valuable if the authors could offer clear explanations that connect the theoretical dynamics to practical real-world implementations.

---

### Decision · Program_Chairs · 2024-10-09

Accept (Poster)